# Selective plasticity of callosal neurons in the adult contralesional cortex following murine traumatic brain injury

Laura Empl[1,2,9], Alexandra Chovsepian[1,2,9], Maryam Chahin[1,2,3], Wing Yin Vanessa Kan[1,2,3], Julie Fourneau[1,2], Valérie Van Steenbergen [1,2], Sanofer Weidinger[1,2], Maite Marcantoni[1,2], Alexander Ghanem[4], Peter Bradley[1,2], Karl Klaus Conzelmann [4], Ruiyao Cai[5,6], Alireza Ghasemigharagoz[5,6], Ali Ertürk[5,6,7], Ingrid Wagner [8], Mario Kreutzfeldt[8], Doron Merkler [8], Sabine Liebscher [1,2,7] & Florence M. Bareyre [1,2,7✉]

Traumatic brain injury (TBI) results in deficits that are often followed by recovery. The contralesional cortex can contribute to this process but how distinct contralesional neurons and circuits respond to injury remains to be determined. To unravel adaptations in the contralesional cortex, we used chronic in vivo two-photon imaging. We observed a general decrease in spine density with concomitant changes in spine dynamics over time. With retrograde co-labeling techniques, we showed that callosal neurons are uniquely affected by and responsive to TBI. To elucidate circuit connectivity, we used monosynaptic rabies tracing, clearing techniques and histology. We demonstrate that contralesional callosal neurons adapt their input circuitry by strengthening ipsilateral connections from pre-connected areas. Finally, functional in vivo two-photon imaging demonstrates that the restoration of pre-synaptic circuitry parallels the restoration of callosal activity patterns. Taken together our study thus delineates how callosal neurons structurally and functionally adapt following a contralateral murine TBI.

[1] Institute of Clinical Neuroimmunology, University Hospital, LMU Munich, 81377 Munich, Germany. [2] Biomedical Center Munich (BMC), Faculty of Medicine, LMU Munich, 82152 Planegg-Martinsried, Germany. [3] Graduate School of Systemic Neurosciences, Ludwig-Maximilians-Universitaet Munich, 82152 Planegg-Martinsried, Germany. [4] Max von Pettenkofer-Institute, Virology, Faculty of Medicine, & Gene Center, LMU Munich, 80336 Munich, Germany. [5] Institute for Tissue Engineering and Regenerative Medicine (iTERM), Helmholtz Zentrum München, Neuherberg, Germany. [6] Institute for Stroke and Dementia Research (ISD), Ludwig-Maximilians-Universität (LMU), Munich, Germany. [7] Munich Cluster of Systems Neurology (SyNergy), 81377 Munich, Germany. [8] Department of Pathology and Immunology, Division of Clinical Pathology, CMU, University & University Hospitals of Geneva, Rue Michel-Servet, 1211 Geneva, Switzerland. [9] These authors contributed equally: Laura Empl, Alexandra Chovsepian. ✉email: florence.bareyre@med.uni-muenchen.de

Moderate traumatic brain injury (TBI) is often followed by regional neuronal cell loss and disruption of neuronal circuits that result in behavioral and cognitive deficits. Both in humans and in animal models, the functional deficits can be followed by spontaneous recovery, suggesting that plasticity of the neuronal circuits disrupted by the injury can compensate for the lost function[1–3]. One region that has been implicated in the recovery process following traumatic or ischemic cortical lesions, is the homotopic contralesional cortex[1,3–10]. Both clinical and experimental studies point to plastic reorganization and changes in neuronal activity levels in the contralesional cortex[9,11–18]. Interestingly, several reports indicate different anatomical sites of plasticity including the formation of new axonal projections from the cortex to neighboring areas or to subcortical areas[6,19–24]. While these observations suggest that the contralesional cortex and its efferent fibers can in principle contribute to recovery[25] or to the emergence of compensatory behavioral strategies[26], it is worth noting that in some settings plasticity and increased activity in the contralesional cortex could also be detrimental and increase injury-mediated deficits[27,28]. These findings indicate that a more refined understanding of the cortical response pattern and in particular of the specific neuronal populations and circuits that adapt to a contralateral lesion is needed.

Recently, it was reported that anatomically connected areas undergo microstructural changes in response to cortical ischemic injuries[16] and that contralesional cortical areas take over the lost function following amputation via adaptive remodeling of callosal inputs[17]. It is thus tempting to speculate that neuronal networks connecting the two hemispheres such as transcallosal connections could play a critical role in the response to cortical injury[29,30]. This is important as, while some higher order information processing, such as attention and language, is lateralized to a region in one hemisphere, correlation of activity between homotopic regions of the two hemispheres is important for sensorimotor processing and hence recovery[31,32]. Thus in this study, we used structural and functional in vivo imaging, selective labeling of neuronal subpopulations as well as circuit mono-synaptic tracing techniques to monitor the circuit rearrangements that take place in the contralesional cortex following TBI. Here, we reveal the selective structural and functional adaptation of callosal neurons and their input circuits to a contralateral brain injury.

functional recovery. Such remodeling processes are likely to take place throughout the CNS and include corticospinal and intraspinal rewiring. Here we wanted to focus on changes that occur in the contralesional cortex so we first examined whether there were apparent changes to the cortical microanatomy. Our results indicated, however, that cortical thickness, neuronal density and soma size contralateral to the lesion were not different between control and injured mice at any time points investigated (Supplementary Fig. 2). As contralesional cortical neurons do neither die nor suffer from atrophy following TBI, we concentrated on analyzing changes to the morphology of individual neurons and in particular their dendritic spines. To do so we used GFP-M mice[36], which sparsely express eGFP in a subset of pyramidal neurons mainly in layer V of the cortex, and performed in vivo two photon imaging of the contralesional sensorimotor cortex before (baselines 1 and 2) and after injury (Fig. 1c). We were able to image and follow 81 dendritic stretches from 4 animals (Fig. 1d) and a total of 10,191 spines throughout the experiment. With two-photon imaging we could show that TBI induced a strong and significant reduction of the spine density that can be detected as early as 3 days following TBI and persisted throughout the study-period (Fig. 1e; ***$p = 0.0001$ B1 vs 3dpi and ***$p < 0.0001$ B1 vs all other time points; one-way repeated measure (RM) Anova followed by Dunnett's test). To assess whether the reduction in spine density was due to a pronounced loss of spines or a lack of newly formed spines, we then followed the fate of the spines and determined the proportion of stable spines, lost spines or newly formed spines (Fig. 1f and Supplementary Fig.3). We found that while the rate of stable spines remained around 80% (Supplementary Fig.3) during the entire study period, the rate of formation and elimination of spines fluctuated over time following injury (Fig. 1f). In particular, the elimination rate of spines was significantly increased compared to the formation rate at 3d post-injury (*$p = 0.0146$ Two-way RM Anova and Tukey test) while the formation rate was significantly increased compared to the elimination rate between days 12 and 18 ($p = 0.0287$ 12–15d and $p = 0.0146$ 15–18d Two-way RM Anova and Tukey test). This argues that adaptive structural plasticity can restore spine density and prompted us to perform additional experiments to determine the neuronal populations involved in these adaptations and the specific time course of these changes following TBI.

## Results

**In vivo two-photon imaging reveals an early general loss of spines and changes in spine dynamics of contralesional cortical neurons following TBI.** We induced TBI unilaterally in the sensorimotor cortex of mice using a controlled cortical impactor (TBI-0310 Impactor, precision Systems and Instrumentations, LLC; Fig.1a) which produced hemicortical lesions that spanned the ipsilateral primary somatosensory cortex (mean lesion volume $7.063 \pm 0.603$ mm$^3$, $n = 4$; Supplementary Fig. 1a) and produced righting reflex and apnea recovery consistent with a moderate injury[33–35] (Supplementary Fig.1b). We first investigated the sensorimotor changes triggered by this brain injury over time using the irregular ladder rung, which allows us to appreciate skilled walking while minimizing the ability of the mice to compensate for impairments through learning. In this test, we observed that acutely following TBI, there is a marked increase in the number of mistakes made by the animals which then recovers over the following weeks (up to day 21 after TBI; Fig. 1b; ***$p = 0.0001$ 3d vs Baseline; *$p = 0.0481$ 7d vs Baseline;#$p = 0.0044$ at 14d, 21d vs 3d;.Friedman test followed by uncorrected Dunn's test). This established that, in our model, remodeling processes in the first few weeks after injury are likely critical for

**The decrease in spine density is specific to callosal neurons.** To determine whether changes in dendritic spine density can be found in all contralesional neurons or are specific to a given population of neurons, we used GFP-M mice and used retrograde tracing with fluorogold. We injected fluorogold into the ipsilesional area prior to TBI to retrogradely label and analyze spine density of callosal neurons located homotopically[37] in the contralesional hemisphere (Fig.2a). Non-callosal neurons (not retrogradely-labeled with fluorogold but whose location was overlapping with the retrogradely-labeled callosal neurons in the contralesional hemisphere) were investigated as controls. We focused on layer II/III and layer V neurons and analyzed spine density in the proximal dendrite, distal dendrite (only for layer V neurons) and apical tufts. We observed that following brain injury dendritic spine density in the apical tufts of callosal neurons is decreased at 7 and 14d (layer II/III 7d $p = 0.0124$, 14d $p = 0.0303$; layer V 14d $p = 0.0329$ One-Way Anova and Dunnett's test) and subsequently recovers to control values by 42d (Fig. 2b–d; left panel). In contrast, dendritic spine density of the apical tufts of non-callosal layer II/III and layer V neurons remained unchanged over the study period (Fig. 2e, f, left panels). We then investigated changes in spine density of proximal

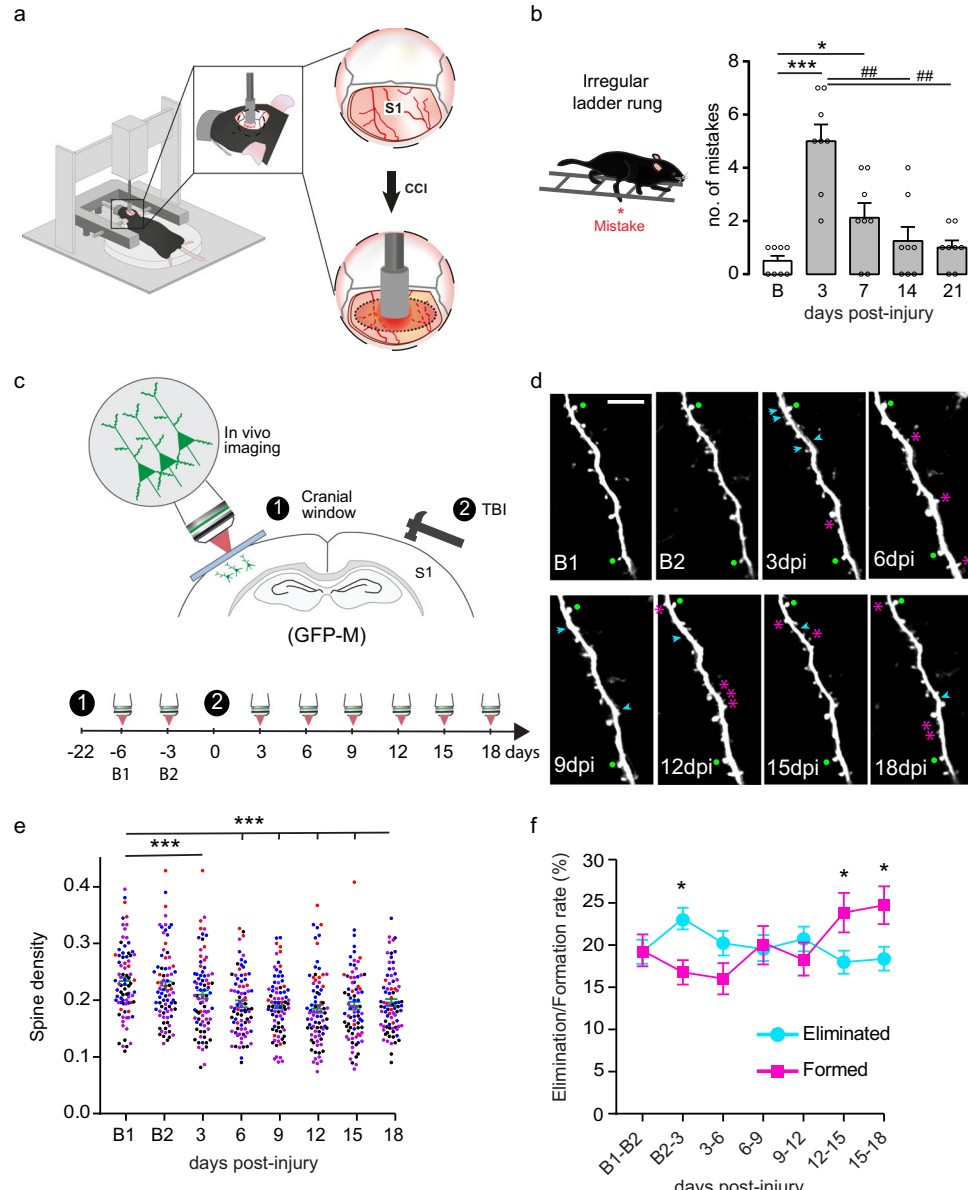

**Fig. 1 Traumatic brain injury triggers a contralesional decrease in spine density and time-dependent variations of the rate of eliminated /formed spines on dendrites of GFP-M mice analyzed using two-photon in vivo imaging. a** Scheme of the moderate brain injury depicting the flat-edged impactor used to lesion the brain (S1: primary somatosensory cortex; CCI: controlled cortical impact). **b** Scheme of the irregular ladder rung test used to evaluate sensorimotor recovery following traumatic brain injury and quantification (mean ± SEM) of the number of mistakes at baseline and following the injury. Data are analyzed with a Friedman test followed by uncorrected Dunn's test. \*\*\*:$p = 0.0001$ 3d vs Baseline; \*:$p = 0.0481$ 7d vs Baseline; ##$p = 0.0044$ at 14d, 21d vs 3d. **c** Scheme of the experimental design and timeline of the two-photon in vivo imaging. **d** Representative timelapse series of an apical dendrite from a layer V neuron in GFP-M mice assessed at the indicated experimental time point and followed up to 18d post-injury. B: Baseline; Scale bar: 10 μm. Cyan arrowhead indicates the position of a lost spine, green dot indicate stable spines present throughout the entire investigational period, and fushia asterisk indicates gained spines. **e** Quantification (mean ± SEM) of spine density at different time points post-injury (dendrites are color-coded per animal). Data are analyzed with one-way RM Anova followed by Dunnett's test. \*\*\*$p = 0.0001$ 3 dpi vs B1; \*\*\*$p < 0.0001$ 6, 9, 12, 15, 18dpi vs B1. **f** Average of elimination and formation rate of spines in GFP-M mice. Data in (**e** and **f**) come from 81 dendritic stretches, 10,191 spines followed over time and $n = 4$ animals. Data are presented as mean ± SEM and analyzed with two-way RM Anova followed by Tukey test. \*$p = 0.0146$ 3 dpi vs B2 eliminated vs formed; \*$p = 0.0287$ 12 dpi vs 15 dpi eliminated vs formed; \*$p = 0.0146$ 15 dpi vs 18 dpi eliminated vs formed. Source data are provided as a Source Data file. B1: Baseline 1; B2: Baseline 2.

dendrites of callosal and non-callosal neurons (Fig. 2b–d, *right panels* and Fig. 2e,f *right panels*). We found a decrease in spine density at 14d in layer II/III callosal neurons ($p = 0.0288$) and a decrease in spine density at 7d ($p = 0.0286$) and 14d ($p = 0.0288$) in layer V callosal neurons while no such changes were observed in layer II/III and layer V non-callosal neurons (One-Way Anova and Dunnett's test for all comparisons). Finally we examined the

spine density of distal dendritic segments of layer V callosal neurons (Fig. 2d, *middle panel*) and could observe a significant albeit transient decrease in spine density at 7d post-injury ($p = 0.0341$ One-Way Anova and Dunnett's test). In contrast, similarly to the apical tufts, no changes in spine density could be seen on the distal compartment of the dendritic trunk of layer V non-callosal neurons (Fig. 2f, *middle panel*). To confirm that the

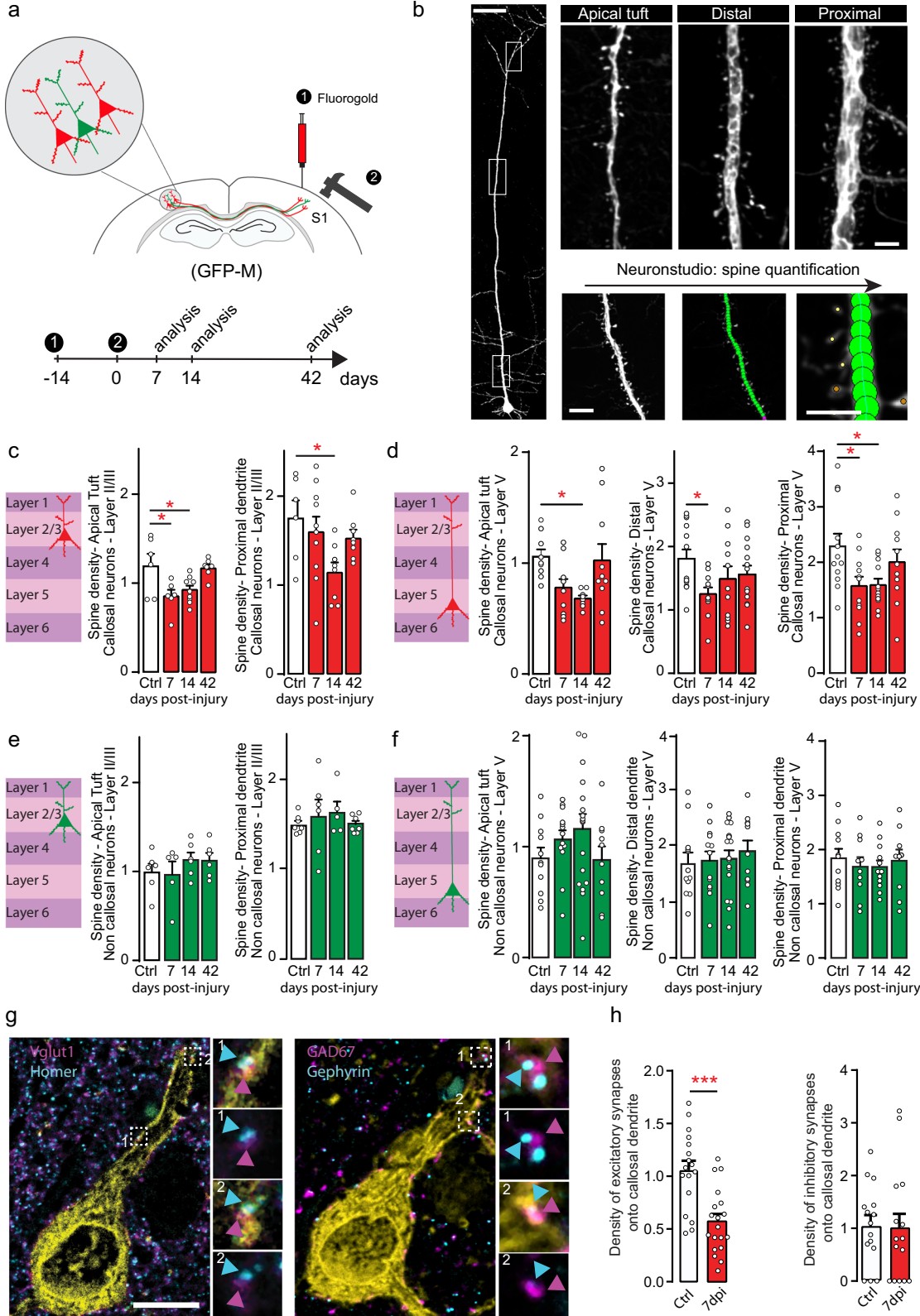

observed changes are indeed selective for neurons that project to the lesioned hemisphere we also examined corticospinal tract (CST) neurons, which were retrogradely labeled from the spinal cord. Here, no differences in spine density could be seen in CST neurons before and 7d after injury (proximal: 2.11 ± 0.22 vs 1.95 ± 0.12; distal: CST 2.1 ± 0.14 vs 2.19 ± 0.12; apical: CST 1.00 ± 0.07 vs 0.96 ± 0.07/μm) or between non-callosal and

corticospinal neurons before and 7d after injury (proximal control: CST 2.11 ± 0.22 vs NC 2.28 ± 0.23; distal control: CST 2.1 ± 0.14 vs NC 1.75 ± 0.15; apical control: CST 1.00 ± 0.07 vs NC 0.94 ± 0.09; proximal 7d: CST 1.95 ± 0.12 vs NC 1.86 ± 0.10; distal 7d: CST 2.19 ± 0.12 vs NC 2.02 ± 0.11; apical 7d: CST 0.96 ± 0.07 vs NC 1.06 ± 0.08/μm). Finally, we investigated how the balance of excitatory and inhibitory inputs onto callosal

**Fig. 2 The decrease in spine density is specific to contralesional neurons directly connected to the lesion site (callosal neurons). a** Scheme of the experimental design and timeline of the confocal ex vivo imaging (red: retrogradely-labelled neurons; green: GFP positive neurons). **b** Confocal images of a layer V cortical neuron (left) and magnification of the dendritic parts boxed in the left images (right) and representative example illustrating the spine quantification using Neuronstudio. Scale bars: 150 μm (left panel), 5 μm (top panels and bottom right panel) and 10 μm (bottom middle panels). **c** Scheme of layer II/III callosal cortical neuron and dendritic spine density (mean ± SEM) of layer II/III callosal population in the apical tuft (left) and proximal dendrite (right) in control (white column) and injured mice (red column) at several time points following injury. $n = 6$–9 independent dendritic stretches from 4 to 5 mice for Apical and $n = 6$–9 independent dendritic stretches from 4 to 5 mice for Proximal. Apical: $p = 0.0124$ and $p = 0.0303$ Ctrl vs 7d and Ctrl vs 14d respectively one-way Anova and Dunnett's test. Proximal: $p = 0.0288$ Ctrl vs 14d one-way Anova and Dunnett's test. **d** Scheme of layer V callosal cortical neuron and dendritic spine density (mean ± SEM) of layer V callosal population in the apical tuft (left), distal dendrite (middle) and proximal dendrite (right) in control (white column) and injured mice (red column) at several time points following injury. $n = 8$–10 independent dendritic stretches from 4 to 5 mice for Apical, $n = 11$–14 independent dendritic stretches from 4 to 5 mice for Distal, $n = 11$–12 independent dendritic stretches from 4 to 5 mice for Proximal. Apical: $p = 0.0329$ Ctrl vs 14d. Distal: $p = 0.0341$ Ctrl vs 7d. Proximal: $p = 0.0286$ Ctrl vs 7d and $p = 0.0288$ Ctrl vs 14d. One-way Anova and Dunnett's test for all. **e** Scheme of layer II/III non callosal cortical neuron and mean ± SEM dendritic spine density of layer II/III non callosal population in the apical tuft (left) and proximal dendrite (right) in control (white column) and injured mice (green column) at several time points following injury. $n = 5$–6 independent dendritic stretches from 4 to 5 mice for Apical and $n = 5$–7 independent dendritic stretches from 4 to 5 mice for Proximal. **f** Scheme of layer V non callosal cortical neuron and dendritic spine density (mean ± SEM) of layer V non callosal population in the apical tuft (left), distal dendrite (middle) and proximal dendrite (right) in control (white column) and injured mice (green column) at several time points following injury. $n = 10$–17 independent dendritic stretches from 4 to 5 mice for Apical, $n = 11$–17 independent dendritic stretches from 4 to 5 mice for Distal, $n = 9$–12 independent dendritic stretches from 4 to 5 mice for Proximal. **g** Confocal images of a retrogradely labeled callosal neuron (Yellow) in the contralesional cortex stained with the excitatory presynaptic marker Vglut1 and the excitatory post-synaptic marker Homer (left panel) and with the inhibitory pre-synaptic marker GAD67 and the inhibitory post-synaptic marker Gephyrin (right panel). Insets show co-localization of pre and post-synaptic marker pairs in contact with the callosal dendrite that we evaluated. Scale bar equals 10 μm. **h** Quantification (mean ± SEM) of the density of excitatory and inhibitory synaptic pairs co-localizing with callosal dendrites. $n = 15$–19 dendritic segments per group from 4 to 5 animal per group. Data are analyzed using unpaired two-tailed $t$-test. $p = 0.0001$ for excitatory synapses and $p = 0.8605$ for inhibitory synapses control (Ctrl) vs TBI. Source data are provided as a Source Data file.

neurons changes following brain injury. To do so, we identified excitatory synapses based on the co-localization of the pre-synaptic marker Vglut1 and the post-synaptic marker Homer and inhibitory synapses based on the co-localization of the pre-synaptic marker GAD67 and the post-synaptic marker Gephyrin on thin paraffin-embedded sections. We found that the density of excitatory inputs but not of inhibitory inputs onto the dendrites of callosal neurons was significantly decreased at 7 days after TBI ($p = 0.0001$ unpaired two-tailed $t$-test; Fig. 2g, h).

**The decrease in spine density is accompanied by alterations of spine morphology**. To further characterize the changes triggered by TBI to contralesional dendritic spines we then characterized the spine morphology as an indicator of the degree of spine maturation. We divided spines into three types: mushroom, thin and stubby spines (Fig. 3a) with mushroom spines generally considered as the most stable and mature structures[38,39]. We investigated spine morphology in the apical tuft of control and injured contralesional callosal layer II/III and layer V neurons in GFP-M mice. We found that, the proportion of the more stable mushroom spines decreased acutely following the injury (layer II/III: $p = 0.0040$ at 7d, $p = 0.0102$ at 14d; layer V: $p = 0.0271$ at 7d and $p = 0.0015$ at 14d; One-Way Anova and Dunnett's test) and partially recovered by 42d in the apical tuft of contralesional callosal neurons. In parallel, the proportion of stubby spines transiently increased at 14d in the apical tufts of layer V neurons ($p = 0.0026$; One-Way Anova and Dunnett's test) before returning to baseline levels at 42d (Fig. 3b,c) arguing for a shortening of the neck. The proportion of thin spines did not change overtime following TBI (Fig. 3b, c). We then investigated spine morphology in the distal and proximal dendrites of callosal neurons in GFP-M mice. Overall, we did not find any changes in the proportion of mushroom, thin and stubby spines following injury in the distal or proximal dendrites of layer II/III and layer V contralesional callosal neurons (Fig. 3b, c). In contrast to callosal neurons, non-callosal neurons do not demonstrate any changes of spine morphology following TBI even on apical tufts (Fig. 3d, e). To complement our spine analysis of contralesional callosal

neurons, we also investigated the response of the callosal axons to injury. To do so, we quantified axons branches of callosal neurons as an indicator of the axonal response to TBI. Both at acute (7d) and chronic (42d) time points, we could see no differences to control animals ($0.34 \pm 0.05$au in control animals vs $0.28 \pm 0.04$au 7d post-injury vs $0.35 \pm 0.05$au 42d post-injury). This suggests that the response of contralesional callosal neurons primarily involves dendritic remodeling with an initial loss and later recovery of pre-synaptic input to those neurons.

**Newly formed spines on callosal neurons after TBI are more stable that those that form on non callosal neurons**. As the spine density of callosal neurons recovers at later timepoints, we wanted to investigate the contribution of spines that form newly after injury to this process. To do so we used two-photon in vivo imaging to track the fate of the newly formed spines of callosal neurons as well as of corticospinal projection neurons (CST neurons), which are located in a similar anatomical location but do not project to the injured cortex. To selectively visualize these neuronal populations, we injected a retrograde adeno-associated virus expressing EGFP (retroAAV-EGFP) either in the ipsilesional cortex (to label callosal neurons, Fig. 4a–c) or in the spinal cord (to label CST neurons, Fig. 4d–f). Using this approach we could follow 23 apical dendrite stretches (8724 spines followed at each time points) for callosal neurons and 13 apical dendrite stretches (4376 spines followed at each time points) for CST neurons. We then analyzed the persistence of pre-existing (spines present before the injury) and newly formed spines and first confirmed that spines that already exist (both on callosal and CST neurons) before the injury are significantly more stable than newly formed spines ($p < 0.0001$; Mann–Whitney test, Fig. 4g). When we specifically compared the stability of the spines that form after injury on callosal neurons to non callosal neurons (CST neurons) we found that the persistence index was significantly higher for newly formed spines on callosal neurons ($p < 0.0001$, Mann–Whitney test, Fig. 4g). This indicates that an increased survival of newly formed spines on callosal neurons can contribute to the recovery of spine density after injury and

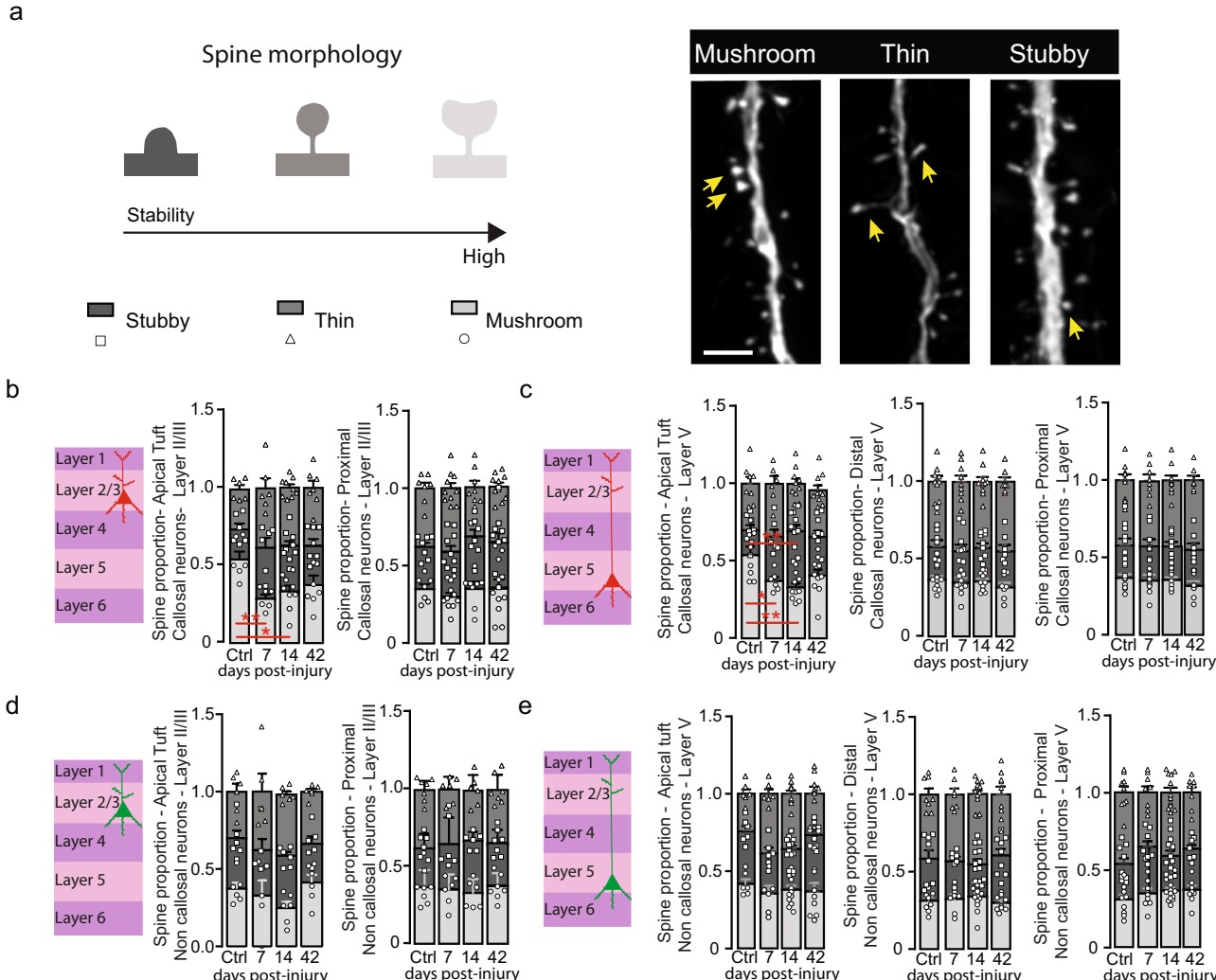

**Fig. 3 Callosal neurons display changes in spine morphology following TBI. a** Schematic representation of spine morphology and confocal images of mushroom, thin and stubby spines. Yellow arrows represent the examples of the respective spine type. Scale bar: 5 μm. **b** Scheme of layer II/III callosal cortical neuron and quantification (mean ± SEM) of the fraction of mushroom, thin and stubby spines on apical tuft and proximal dendrites of layer II/III callosal neurons. $n = 6$–9 independent dendritic stretches from 4 to 5 mice for Apical and $n = 7$–10 independent dendritic stretches from 4 to 5 mice for Proximal. Apical mushroom spines: $p = 0.0040$ Ctrl vs7d and $p = 0.0102$ Ctrl vs 14d. One way-ANOVA and Dunnett's test. **c** Scheme of layer V callosal cortical neuron and quantification (mean ± SEM) of the fraction of mushroom, thin and stubby spines on apical tuft, distal and proximal dendrites of layer V callosal neurons. $n = 7$–10 independent dendritic stretches from 4 to 5 mice for Apical, $n = 7$–10 independent dendritic stretches from 4 to 5 mice for Distal, $n = 6$–10 independent dendritic stretches from 4 to 5 mice for Proximal. Apical mushroom spines: $p = 0.0271$ Ctrl vs7d and $p = 0.0015$ Ctrl vs 14d. One way-Anova and Dunnett's test. Apical stubby spines: $p = 0.0026$ Ctrl vs 14d. One way-Anova and Dunnett's test. **d** Scheme of layer II/III non callosal cortical neuron and quantification (mean ± SEM) of the fraction of mushroom, thin and stubby spines on apical tuft and proximal dendrites of layer II/III non callosal neurons. $n = 5$–6 independent dendritic stretches from 4 to 5 mice for Apical and $n = 6$–7 independent dendritic stretches from 4 to 5 mice for Proximal. **e** Scheme of layer V non callosal cortical neuron and quantification (mean ± SEM) of the fraction of mushroom, thin and stubby spines on apical tuft, distal and proximal dendrites of layer V non callosal neurons. (light gray: mushroom spines, dark gray: stubby spines, medium gray: thin spines). $n = 6$–12 independent dendritic stretches from 4 to 5 mice for Apical, $n = 7$–13 independent dendritic stretches from 4 to 5 mice for Distal, $n = 8$–12 independent dendritic stretches from 4 to 5 mice for Proximal. Statistical $p$ values are always compared to control (Ctrl) values. Source data are provided as a Source Data file.

underlines the functional significance of the associated rewiring of the callosal input circuitry.

**Callosal neurons adapt their input circuitry by re-establishing ipsilateral connections from pre-connected areas following TBI.** The observation that callosal neurons can remodel their input connections prompted us to study how the input circuitry of callosal neurons evolves after TBI. To answer this question, we first implemented retrograde mono-synaptic tracing using rabies virus[40,41] (Fig. 5a) combined with tissue clearing[42] (Fig. 5b).

We first verified that the mono-synaptic tracing was specific and does not show any leakage by injecting either the rabies virus in absence of the complementing G and EnvA proteins as well as the cre recombinase or either injecting the AAV expressing the envA and G proteins (AAV1-synP-DIO-sTpEpB-eGFP) without rabies virus. Trans-synaptic retrograde tracing and tissue clearing allowed us to identify the general location of presynaptic inputs of callosal neurons before and following injury (Fig. 5c). This approach allowed us to determine that the major presynaptic input of contralesional callosal neurons is located in the

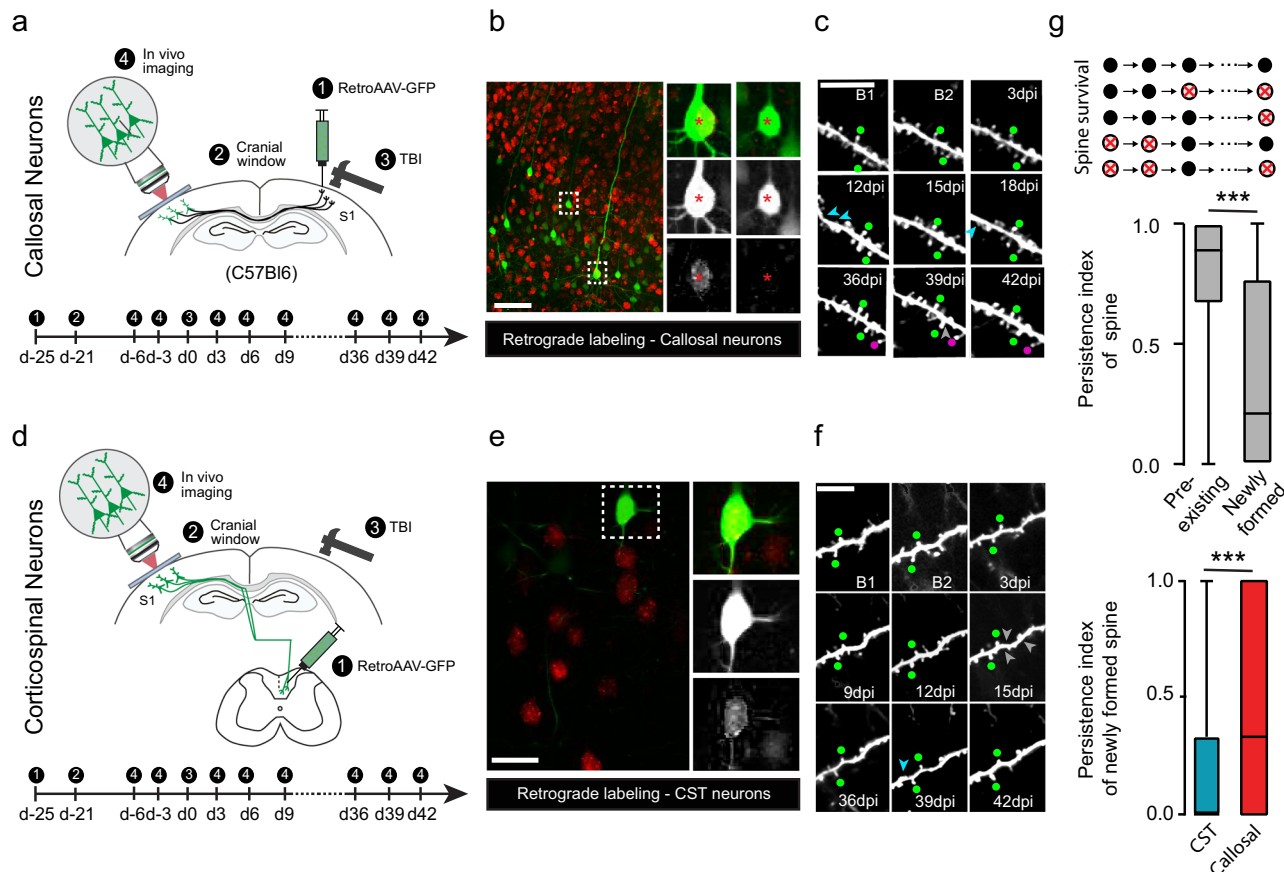

**Fig. 4 Differential stability of newly generated spines formed on callosal and non callosal neurons following traumatic brain injury. a** Scheme of the experimental design with retroAAV injected in the lesion area and timeline of the two-photon in vivo imaging. **b** Confocal images of retrogradely labeled callosal neurons in the cortex (Green: retroAAV-EGFP; red: NT435). Insets are four times magnified (top raw overlay; middle raw: retroAAV-EGFP; bottom raw: NT435).Scale bar equals 50 μm. **c** Representative timelapse series of an apical dendrite from a callosal neuron retrogradely-labeled with retroAAV-EGFP. Each image shows the same dendrite stretch at a specific experimental time point before and up to 42d post-injury (23 independent dentritic stretches and 8724 spines were followed). Cyan arrowheads indicate disappearing pre-existing spines. Gray arrowheads indicate disappearing newly-formed spines. Green dot indicate stable pre-existing spines. Magenta dots indicate stable newly-formed spines. B1: Baseline 1; B2: Baseline 2; Scale bar: 10 μm. **d** Scheme of the experimental design with the retroAAV injected in the spinal cord and timeline of the two-photon in vivo imaging. **e** Confocal images of retrogradely labeled CST neurons in the cortex (Green: retroAAV-EGFP; red: NT435). Insets are two times magnified (top raw overlay; middle raw:retroAAV-EGFP; bottom raw: NT435). Scale bar equals 50 μm. **f** Representative timelapse series of an apical dendrite from a CST neuron retrogradely-labeled with retroAAV-EGFP. Each image shows the same dendrite stretch at a specific experimental time point before and up to 42d post-injury (13 independent dentritic stretches and 4376 spines were followed). Cyan arrowheads indicate disappearing pre-existing spines. Gray arrowheads indicate disappearing newly-formed spines. Green dot indicate stable pre-existing spines. Magenta dots indicate stable newly-formed spines. B1: Baseline 1; B2: Baseline 2; Scale bar: 10 μm. **g** Quantifications comparing the persistence of all pre-existing spines ($n = 377$ spines analyzed) to all newly formed spines ($n = 187$ spines analyzed) (top panel) and the persistence of newly formed spines on callosal ($n = 134$ spines from nine dentritic stretches from four animals) and CST neurons ($n = 55$ spines analyzed from eight dendritic stretches from four animals) (bottom panel). All data (top and bottom) are analyzed using two-tailed Mann–Whitney test ($p < 0.0001$ in both cases) and presented as box plots (top panel: Pre-existing spines: minima:0, maxima:1,median = 0.8889; 25% percentile = 0.6667; 75% percentile = 1; newly formed spines: minima:0, maxima:1; median = 0.2111; 25% percentile = 0; 75% percentile = 0.7708 – bottom panel: CST: minima:0, maxima:1, median = 0; 25% percentile = 0; 75% percentile = 0.3333; Callosal: minima:0, maxima:1,median = 0.3333; 25% percentile = 0; 75% percentile = 1) Source data are provided as a Source Data file.

contralesional somatosensory cortex and showed that presynaptic cortical areas that are strongly connected were largely similar between control and TBI mice, with only subtle changes (Supplementary movie and Fig. 5c). As only presynaptic cells could be visualized and starter cells could not be detected using clearing techniques, we complemented this analysis using conventional immunohistological approaches. We sectioned, stained and analyzed the entire mouse brain and obtained connectivity ratios by normalizing the counted number of presynaptic cells to the number of starter cells (Fig.5d, e). Starter neurons (mCherry + / GFP + ) were located exclusively within the injected area in the somatosensory cortex (Fig. 5e) and were surrounded by many

mCherry + -only, monosynaptic input neurons (Fig. 5e). We analyzed more than 20 brain regions and identified the areas that provide input to callosal neurons in controls, in both ipsilesional and contralesional cortices (Fig. 5f), with the somatosensory cortex exhibiting the highest connectivity ratio. Other cortical inter-areal connectivity included afferents from the motor cortex, from the auditory area, the retrosplenial area or the posterior parietal association area, among others (Fig. 5f). Subcortical inter-areal connectivity included mostly the thalamus and at much lower ratios the hypothalamus and caudate putamen (Fig. 5f). Importantly, all of these regions have been found to project to the somatosensory cortex before[43] and we detected no aberrant input.

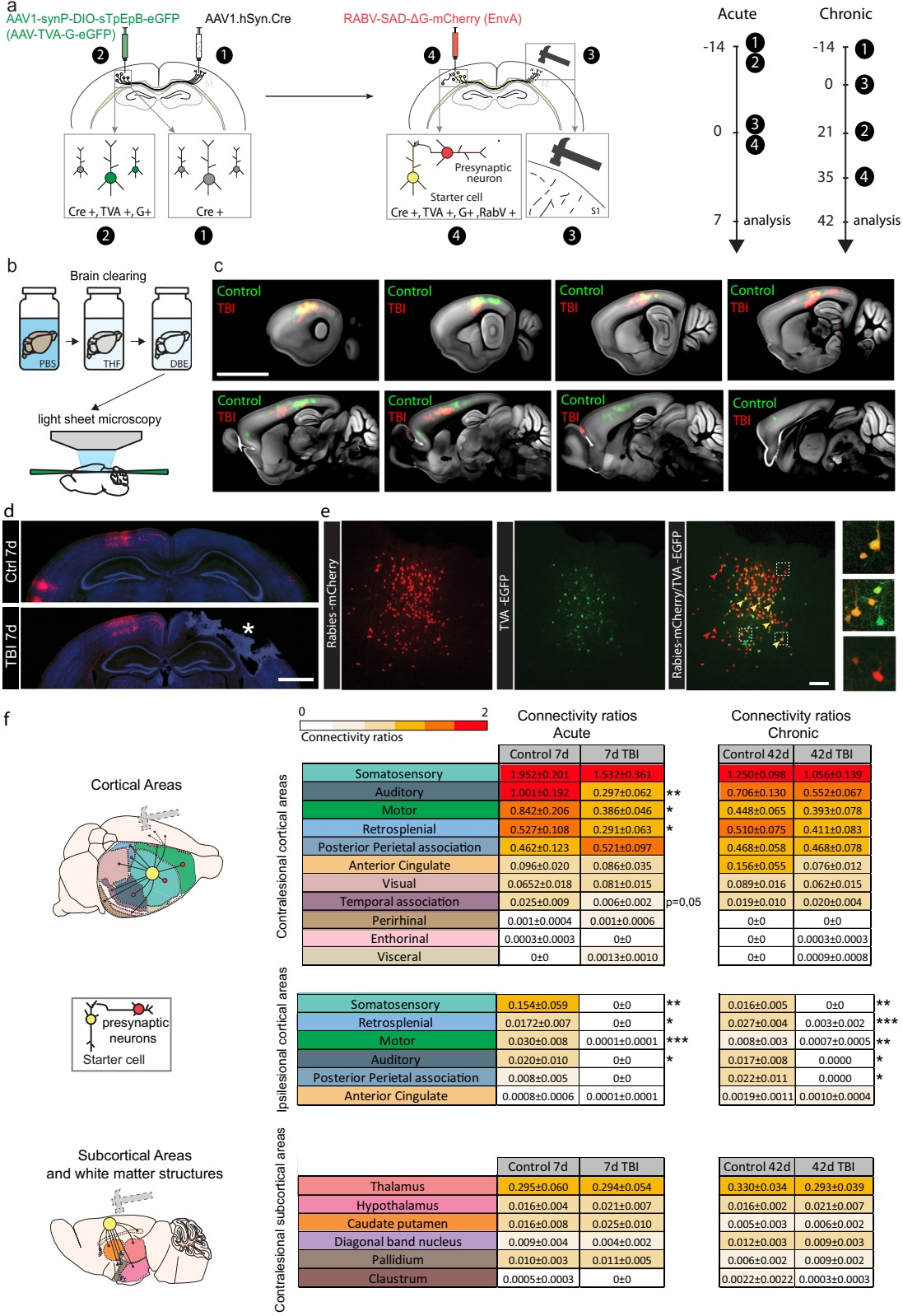

Acutely (7d) following the lesion we observed a general reduction in connectivity within the contralesional somatosensory area, motor area ($p = 0.0419$), retrosplenial area ($p = 0.0405$), the auditory cortex ($p = 0.0021$) and temporal association area ($p = 0.0503$), in particular (all two-tailed $t$-test, Fig. 5f). Additional contralesional areas, albeit whose connectivity ratio was lower initially showed no significant change (anterior cingulate

area, visual and perirhinal cortices). All areas located in the ipsilesional cortex fully dropped in connectivity, as expected, as this is the site of the brain injury. At chronic time points, 42d post TBI, these ispilesional areas remained low in connectivity. Remarkably, during the chronic phase following TBI, we observed a pronounced recovery of the connectivity in the contralesional hemisphere with a normalization of input from the

**Fig. 5 Circuit connectivity is adaptive following traumatic brain injury. a** Scheme of trans-synaptic circuit tracing, clearing and experimental timelines for acute and chronic connectivity. **b** Schematic of brain clearing and light-sheet fluorescence microscopy. **c** Serial light-sheet fluorescence microscopy images of cleared brain demonstrating the location of presynapstic neurons in control (green) and injured (red) mice. Scale bar: 5 mm. **d** Representative images of Rabies labeling in the cortex (red: presynaptic cells). Scale bar: 1 mm. **e** Confocal images of starter cells and presynaptic cells in the cortex (green: neurons infected with TVA-GFP; red: neurons infected with rabies-mcherry and yellow: starter neurons). Scale bar: 100 μm. **f** Scheme of the connectivity between the somatosensory cortex and contralesional cortical (upper) and subcortical (lower) areas and tables of contralesional and ipsilesional connectivity ratios (mean ± SEM) in controls 7d, controls 42d, and TBI 7d and TBI 42d. The connectivity ratios are color-coded and significance from matching controls is reported. All data were analyzed with two-tailed $t$-test. Auditory: $p = 0.0021$ Ctrl 7d vs TBI 7d; Motor: $p = 0.0419$ Ctrl 7d vs TBI 7d; Retrosplenial: $p = 0.0405$ Ctrl 7d vs TBI7d; Temporal association: $p = 0.0503$ Ctrl 7d vs TBI 7d; Ipsilateral somatosensory: $p = 0.0047$ Ctrl 7d vs TBI 7d; $p = 0.0024$ Ctrl 42d vs TBI 42d; Ipsilateral Retrosplenial: $p = 0.02$ Ctrl 7d vs TBI 7d; $p < 0.0001$ Ctrl 42d vs TBI 42d; Ipsilateral Motor: $p = 0.0008$ Ctrl 7d vs TBI 7d; $p = 0.0019$ Ctrl 42d vs TBI 42d; Ipsilateral Auditory: $p = 0.0492$ Ctrl 7d vs TBI 7d; $p = 0.0152$ Ctrl 42d vs TBI 42d; Ispilateral Posterior Perietal association:$p = 0.0294$ Ctrl 42d vs TBI 42d. $n = 6$–12 regions of interest from 6 to 8 independent animals per group. Source data are provided as a Source Data file.

motor, auditory, retrosplenial and temporal association areas (all two-tailed $t$-test, Fig. 5f) when injured animals at 42d were compared to their matching controls. Subcortical areas, which had originally low connectivity ratios with the examined contralesional somatosensory cortex, demonstrated relatively minor changes both acutely and chronically following TBI (Fig. 5f). Our in situ analysis therefore underscores that callosal neurons restore their pre-synaptic connectivity largely by strengthening ipsilateral connections from pre-connected areas and is therefore largely homeostatic in nature, thereby restoring hitherto lost connections.

**TBI causes a widespread, transient reduction in neuronal activity in the contralesional cortex.** As we have shown that TBI causes dendritic remodeling of callosal neurons that include the loss of excitatory inputs early after lesion, we then asked whether these changes might also translate into alterations of neuronal function. To address this question, we longitudinally assessed neuronal activity in vivo using two-photon calcium imaging using the genetically encoded calcium indicator GCaMP6m in callosal and non-callosal neurons in the contralesional cortex (Fig. 6a, b). Callosal neurons were retrogradely labelled by the injection of retroAAV-tdTomato into the lesion area prior to brain trauma (Fig. 6a). At the level of the proportion of active cells a clear decrease was detected 10 days after the injury for both neuronal populations (Fig. 6c, baseline (B) vs 10d: callosal neurons $p = 0.029$; non-callosal neurons $p = 0.015$, two-tailed $t$-test). The fraction of active cells normalized after 42d both in callosal as well as in non-callosal neurons (Fig. 6c, 10d vs 42d: callosal neurons $p = 0.027$, non-callosal $p = 0.047$, two-tailed $t$-test) and were not significantly different from the baseline recording anymore. When we followed the same neurons over time we found that TBI also caused a pronounced reduction of neuronal activity in both callosal and non-callosal neurons 10 days after injury (Fig. 6d, baseline vs 10d: callosal neurons $p = 0.013$, non-callosal neurons $p < 10^{-10}$, KS test). After 42 days we observed a recovery in activity levels primarily in non-callosal neurons (10d vs 42d: callosal neurons $p = 0.23$, non-callosal neurons $p = 0.012$, KS test). The activity levels, however, were still reduced compared to the baseline recordings in non-callosal ($p < 10^{-4}$, KS test). More specifically, the activity of cells active at baseline was strongly reduced 42 days after injury for both non-callosal and callosal neurons (Fig. 6e, callosal neurons $p < 10^{-5}$, non-callosal neurons $p < 10^{-12}$, two-tailed $t$-test). The initial activity levels were also determining the likelihood of remaining active until at least 42 days after the injury. Neurons with a baseline activity level of ≥1 transient per minute ('high') were significantly more likely to remain active as opposed to neurons with a lower activity level (<1 transient/min, 'low') (Fig. 6f, callosal neurons $p < 0.05$, non-callosal neurons $p < 0.05$, nonparametric bootstrapping).

We further investigated the fate of neurons initially active at baseline. This investigation revealed that a larger fraction of callosal compared to non-callosal neurons went silent at 10d after TBI ($p < 0.05$, nonparametric bootstrapping), while the fraction of persistently active neurons or neurons that regained activity after 42d, or those turning silent only at d42 did not differ between the two populations (Fig. 6g). When investigating the history of cells active at day 42, we found that a large majority of these cells only became active after TBI ('newly active') both in callosal as well as in non-callosal neurons (Fig. 6h). Only ~20% of cells active at day 42 were persistently active cells, while 24 and 29%, in callosal and non-callosal neurons regained activity after turning silent at day 10 ($p = $ ns; Fig. 6h).

## Discussion
The adaptation of contralesional cortex following brain injury is an important paradigm that has been studied to understand how the adult brain responds to damage. However, if and how such adaptive responses contribute to functional recovery is not entirely resolved[26,28] and the specific neuronal populations and circuits that underlie these responses have not been identified. In this study, we used a combination of techniques ranging from retrograde viral labeling, in vivo timelapse two-photon imaging of spine turnover, trans-synaptic circuit tracing, tissue clearing and in vivo calcium imaging to reveal the specific adaptation of contralesional cortical callosal neurons directly connected to the injury site following TBI (Table 1). We find, using two-photon longitudinal imaging over 18 days that there is a strong early decrease in spine density over at least 2 weeks with signs of recovery thereafter. The plasticity of contralesional areas following brain injury has been established previously thanks to observations in patients[44–46] and in animal models[47–51]. In particular changes in neurotransmitters[52], dendritic growth and synapse formation have been described[53,54]. Reports also point out changes in neuronal activity in the contralesional cortex that are correlated with functional impairments in the acute and subacute phases following injury[10,18,55–57]. In our study, we find, among others, that callosal neurons, which homotopically connect both cortical hemispheres, show a unique vulnerability and adaptation following TBI with a homeostatic reintegration into existing circuits. Recent clinical observations reported a specific vulnerability of remote brain regions directly connected to ischemic brain lesion[16,58]. Likewise, recent experimental work indicates a specific and profound plasticity of callosal input following peripheral injury that underlies cortical takeover[17]. In line with this view, our study now reveals the selective adaptation of callosal neurons and their input circuits after a contralateral brain injury. Interestingly, clinical observations following visual cortical damage that leads to blindness also point toward a similar compensatory mechanism in which areas in the visual cortex of

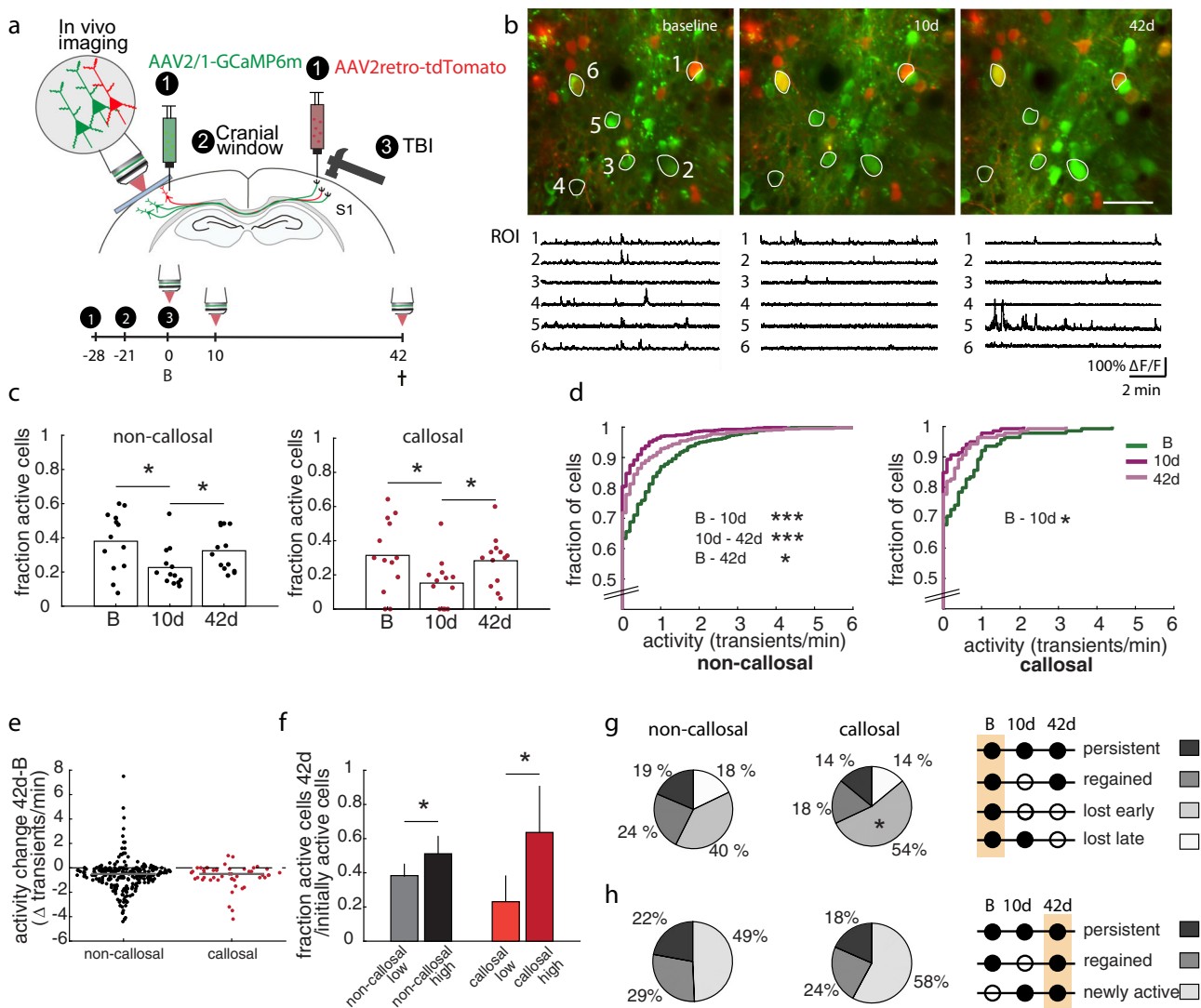

**Fig. 6 Contralesional neurons display dynamic alterations of spontaneous activity levels following traumatic brain injury. a** Scheme of functional in vivo imaging experiments investigating contralesional neurons using AAV-GCaMP6m. **b** Average projection of a representative field of view (FOV) showing neurons transduced with AAV-GCaMP6m in the contralesional hemisphere (green) and retrogradely transduced with retroAAV-CAG-tdtomato (red) at different time points following traumatic brain injury (B – baseline). Bottom: Calcium traces of example neurons marked within the FOV images. Scale bar equals 50 μm. **c** Fraction of active cells at baseline and two different time points following traumatic brain injury for callosal and non-callosal cells. $n = 13$ independent fields of views (663 non callosal and 139 callosal neurons) from six mice. Baseline (B) vs 10d: callosal neurons $p = 0.029$; non-callosal neurons $p = 0.015$, two-tailed $t$-test. 10d vs 42d: callosal neurons $p = 0.027$, non-callosal $p = 0.047$, two-tailed $t$-test. **d** Cumulative distributions of calcium transient frequencies at the indicated time points following traumatic brain injury. Note a shift in the distribution toward lower frequencies at 10d following lesion and a slight recovery at 42d. Baseline vs 10d: callosal neurons $p = 0.013$, non-callosal neurons $p < 10^{-10}$, KS test). 10d vs 42d: callosal neurons $p = 0.23$, non-callosal neurons $p = 0.012$, KS test. Baseline vs 42d non-callosal $p < 10^{-4}$, KS test. **e** Activity change between baseline and 42d in initially active non-callosal and callosal neurons. **f** Fraction of active cells at 42d dependent on the initial activity levels at baseline in non-callosal and callosal neurons (≥1 transient/min: 'high', <1 transient/min: 'low'). Note that the initial activity levels determines the likelihood of remaining active at 42 days after the injury. Data analyzed with non-parametric bootstrapping of 10,000 iterations. Error bars represent the 95% confidence interval (CI). **g** Fate of non-callosal and callosal neurons active at baseline. **h** Activity history of non-callosal and callosal neurons active at 42d., *$p < 0.05$, **$p < 0.01$, ***$p < 0.001$. Source data are provided as a Source Data file.

the contralesional hemisphere are recruited to compensate for altered visual function[59]. One aspect of the adaptation of callosal neurons is that the spines that are lost early after the TBI lesion are replaced over time with newly formed spines that on average persist longer than the spines that are formed over the same time period in non-callosal neurons. This underlines the functional significance of callosal input circuitry plasticity. Such a re-establishment of stable spines has previously been described in response to sensory deprivation in mice[60] and might be an attempt to re-establish a response to spared inputs or to recruit

new inputs. Notably the spine morphology of callosal neurons evolves in line with the changes of spine density with more immature spines appearing at acute time points after injury and more mature and stable spine morphologies emerging later on. This is interesting as it suggests a biphasic plastic and adaptive response of callosal spines after injury. It therefore demonstrates that connected neurons in the contralesional cortex are uniquely affected by and respond to the injury in specific ways. This is in line with recent results showing a specific adaptation of callosal connections to peripheral injury[17]. It is important to note, in

**Table. 1 Summary table of the changes and adaptations of callosal neurons following contralateral traumatic brain injury.**

| Manipulation | Acute time points | Later time points | Figure panel |
|---|---|---|---|
| Sensorimotor function | Decreased | Normalized | 1b |
| Spine density in the contralesional cortex | Decreased | _ | 1e |
| Spine density of callosal neurons | Decreased | Normalized | 2c & d |
| Spine density of non callosal neurons | Not changed | Not changed | 2e & f |
| Excitatory input onto callosal neurons | Decreased | _ | 2g & h |
| Inhibitory input onto callosal neurons | No changed | _ | 2g& h |
| Spine morphology of callosal neurons | Altered | Normalized | 3b & c |
| Spine morphology of non callosal neurons | Not changed | Not changed | 3d & e |
| Persistence of new spines on callosal neurons | _ | Higher in callosal neurons compared to control neurons | 4g |
| Connectivity ratios of callosal neurons | Decreased | Normalized | 5 |
| Activity patterns callosal neurons | Decreased | Normalized | 6c |
| Activity patterns non callosal neurons | Decreased | Normalized | 6c |

addition, that the changes triggered in the contralesional hemisphere can depend on the lesion extent as previous studies have shown that the balance between adaptive and mal-adaptive responses are affected by the size of the initial brain injury. It is therefore quite possible that the structural and functional alterations of callosal neurons that we observe here as well as their contribution to recovery can be affected by the type and size of the brain injury and could well be different in case of smaller or bigger lesion sizes[26,27,61] or for lesions induced in the developing or aged CNS[58]. As the TBI destroys axonal projections of the contralesional callosal neurons, we reasoned that plasticity at the level of those axons could also occur as a response to the injury and therefore investigated this possibility. It is known that during development callosal neurons extend profuse axonal projections till about post-natal day 8, and these connections are then refined[62,63] in an activity-dependent process till p21 to obtain the adult pattern of cross-hemispheric homotopy[64]. Here we have found no additional branching of callosal axons neither at acute nor at chronic time points following TBI indicating that the plastic processes could be restricted to the dendritic remodelings in the contralesional hemisphere.

As callosal neurons are uniquely affected by and responsive to TBI, we wondered whether and how their input circuitry would re-arrange at acute and chronic time points following injury. To answer this question we made use of the rabies virus, which monosynaptically labels pre-synaptic neurons that project to callosal neurons. We took a two-step approach: we first identified the brain regions connected to contralesional callosal neurons using tissue clearing and large scale imaging. Next we then determined the connectivity ratios before and after brain injury of the connected areas identified in tissue clearing analysis. Not surprisingly, we find that the main input to the somatosensory cortex is provided by cortical areas and in particular the somatosensory cortex, the motor cortex and the auditory cortex. This is in agreement with previous studies[43,65]. We further observed remarkably little pre-synaptic input at 42d to ectopic cortical or subcortical regions, as most brain regions connected before the injury were also found to provide input at 42d. This is important as restored connectivity to somatomotory areas such as the primary or secondary somatosensory cortex and primary motor cortex might underlie favorable rather than mal-adaptive adaptations[17]. Finally, we sought out to understand the state of activity of those callosal neurons and performed functional imaging experiments using genetically encoded calcium sensors. Functional imaging revealed that the fraction of active callosal neurons initially decreased but recovered over time. This initial decrease in neuronal activity in callosal neurons is in line with the synaptic structural changes seen in those neurons early following

the injury. Indeed, we demonstrate that concomitantly to the decrease in activity, callosal neurons also demonstrate a specific loss of excitatory input onto their dendrite. This structural loss of excitatory input is consistent with the functional decrease in neuronal activity also observed in other paradigms[66,67]. In addition, while the callosal neurons active at baseline mostly fail to maintain and regain activity, a large part of the activity seen at 42d is due to newly active cells. This probably reflects activity-dependent functional refinements as circuit plasticity is correlated with increased spine dynamics[43,68], which we observe during this period. In line with this, monosynaptic tracing revealed that input in discrete regions increases from 7 to 42 dpi also, supporting the notion of ongoing synapse refinement. Our functional imaging data also show that callosal neurons did not behave fundamentally differently from non-callosal neurons. Why callosal and non-callosal neurons behave similarly can probably be explained by intracortical communication, as non-callosal neurons receive input from callosal neurons and vice versa.

Taken together, our study reveals important structural and functional principles of remodeling following TBI: it identifies a specific neuronal population (callosal neurons) that are particularly responsive to contralateral lesions and are thus prime targets for therapeutic interventions. Considering the importance of cortical remodeling processes for the functional recovery not only after traumatic but also after ischemic nervous system injuries, this has important implications for many common neurological conditions.

## Methods
All experiments were approved by the Regierung von Oberbayern under the protocol number AZ:55.2.1.54-2532.135-15.

**Animals.** GFP-M mice (24 mixed gender randomized equally per group) and 78 female C57Bl6J mice, 8–12 weeks-old, were used for this study. All animals were housed under controlled standard housing conditions (dark/light cycle of 12 h, temperature 22 ± 2 degrees and humidity of 55 ± 10%) with food and water ad libitum.

**Controlled cortical impact (CCI).** Mice were anaesthetized via an intraperitoneal (i.p.) injection of MMF (Medetomidin 0.5 mg/kg, Orion Pharma; Midazolam 5.0 mg/kg, Ratiopharm; Fentanyl 0.05 mg/kg, B.Braun). When pedal reflex disappeared, animals were fixed on the stereotactic frame (Precision Systems & Instrumentation, LLC). The skin was incised and a square craniotomy (4 mm × 4 mm), positioned between Lambda, Bregma, the sagittal suture and the side of the skull was made. The injury to the brain was induced using a TBI-0310 impactor (Precision Systems & Instrumentation, LLC), in the area of the somatosensory cortex (flat-edged rod of 3 mm diameter, velocity of 6 m/s, a dwell time of 150 m/s and a depth of 0.3–0.5 mm). The removed bone piece was re-positioned to its previous location and glued to the skull with Vetbond (3 M™ Vetbond™, 3 M United States). The skin was sutured. The mean apnea duration and righting reflex

duration (time for a mouse to flip onto its feet from a supine position after the injury) were recorded in order to evaluate the severity of the injury.

**Evaluation of sensorimotor function.** *Ladder rung test:* For assessment of fine paw placement, the ladder rung test was used[69]. In this test, the animals had to cross a 1 m horizontal grid ladder and mistakes were counted by an investigator blinded to time points based on video recordings frame-by-frame of three consecutive crossings. We evaluated the animal's ability for fine coordinated paw placements using irregular spacing of the rungs (irregular walk task). Only consecutive steps of the hindlimbs were analyzed. Therefore, the last step before or after any interruptions were not scored. Placements were considered as a mistake when mice either totally missed a rung or if they slipped from a rung (deep or slight slip). Placements were considered as correct when the mice correctly placed all the foot or only a portion of the foot on the rungs. Then the number of slips over a standard distance was calculated quantitatively.

**Evaluation of neuronal density, soma size, spine density, spine morphology and synaptic input onto callosal dendrites in fixed tissue**
*Tracer injections.* For experiments on fixed tissue imaged with the confocal microscope, we retrogradely labeled callosal projection neurons by stereotactically injecting 0.3 μl of FluoroGold; (1% in 0.1 M Cacodylate buffer, Fluorochrome LLC) in the ispilesional somatosensory cortex (coordinates from Bregma: rostrocaudal −1.5 mm, lateral 1.7 mm, depth 0.3 mm and timeline is depicted in Fig. 2). In short, a glass capillary micropipette tip was slowly inserted into the brain through a small hole drilled in the skull. In order to avoid backflow, the pipette remained in the brain for a minimum of three minutes after injection completion[70]. To retrogradely label corticospinal (CST) projection neurons, laminectomy was performed at thoracic level 8 of the spinal cord and 0.5 μl of FluoroGold were stereotactically injected with a glass capillary into the right side of the spinal cord, 2 mm lateral from spinal midline at a depth 0.3 mm. The micropipette remained in place for at least 3 min after completing the injection to minimize backflow. Subsequently, the muscles and skin were sutured and saline was administered subcutaneously. For the experiment aiming at observing the changes in synaptic input onto callosal dendrites, we injected the somatosensory cortex with 0.5 μl a retrograde AAV expressing tdTomato (Retro-AAV-tdTomato, Addgene #59462) at the following coordinates from Bregma (rostrocaudal −1.5/lateral 1.7/deepth 0.3 mm).

*Tissue preparation for spine imaging.* Mice were sacrificed at 7, 14, and 42 days post-injury or at 12 days after injection (control non-injured group) and perfused with 4% paraformaldehyde (PFA) in 0.1 M PBS. Brains were kept in 4% PFA for 24 h, dissected, immersed in 3% low-melting-point agarose and cut with a Leica VT 1000 s vibratome. Coronal sections (100 μm) were cut and washed three times with 1× PBS for 10 min before free-floating staining with NeuroTrace 435 (Thermo Fischer Scientific; 1:500 in 0.1% Triton PBS) at 4 °C, overnight. The next day sections were washed three times in 1 × PBS for 10 min and mounted on gelatin subbed slides, using VectaShield (Vector Laboratories).

*Tissue preparation for synaptic input onto callosal dendrite quantification.* For thin paraffin sections, mouse brains were paraffine embedded and cut into 2 um thin sections. After deparaffination, antigen retrieval was performed using a Pascal Citrate buffer at pH6.0. To inactivate endogenous peroxidases, slides were incubated with Dako REAL peroxidase-blocking solution (Dako, K0672) and subsequently blocked with 10% FCS in PBS.

*Immunohistochemistry.* To quantify neuronal density and soma size, we stained free-floating (Jacobi et al., 2015). To do so, the sections were washed for 10 min in 1× PBS three times followed by 1 h incubation in blocking buffer (1× PBS with 0,5% Triton and 10% goat serum) at room temperature. Brain sections were then incubated overnight at 4 °C in the primary antibody solution: 1× PBS containing 0.1% Triton, 1% goat serum and mouse anti-NeuN Ab (1:500; Thermo Fischer Scientific). The next day the tissue was washed three times for 10 min in 1× PBS 0.1% Triton before the application of the secondary antibody solution goat anti-mouse AlexaFluor® 594 Ab (1:500; Invitrogen) in 1× PBS −0.1% Triton, 1% goat serum. After overnight incubation at 4 °C the tissue was again washed three times for 10 min in 1× PBS, mounted on gelatin coated glass slides in VectaShield (Vector Labs). To quantify synaptic input onto callosal dendrites in thin paraffin-sections (2 μm thin) we first stained retrogradely-labeled callosal neurons with rabbit anti-RFP antibody (1:400; abcam, ab124754) diluted in Dako REAL antibody diluent (Dako, S2022) at 4 °C o/n. To visualize the specific signal, anti-rabbit HRP (Dako,K4003) together with Opal-570 (1:600; Akoya, FP1488001KT) was used as secondary antibody and amplification system. After washing, slides were incubated with goat-anti-mouse IgG Fab-fragments (1:100; Jackson ImmunoResearch, 115-007-003) and goat serum to avoid subsequent unspecific binding. To perform the synaptic staining the sections were incubated with mouse IgG2b anti-Gephyrin antibody (1:200; Santa Cruz, sc-25311) and mouse IgG1 anti-GAD65/67 antibody (1:100; Santa Cruz, sc-365180) or chicken anti-Homer1 antibody (1:500; Synaptic System, 1600006) and mouse IgG1 anti- BNPI antibody (1:100; Santa Cruz, sc-377425), respectively. Conjugated secondary antibodies corresponding to the

species and isotype of the primary antibody-pairs were used to visualize specific staining. Nuclei were stained with DAPI (Invitrogen, D1306). Slides were mounted with Fluoromount aqueous mounting medium (Sigma-Aldrich, F4680).

*Image acquisition and image processing.* Callosal neurons were identified as GFP and Fluorogold positive, while GFP-positive but Fluorogold-negative neurons located in the same area than callosal neurons were considered non-callosal. CST-neurons were identified as double positive neurons for Fluorogold and GFP.

*Cortical thickness, cell density and cell size.* To measure the cortical thickness, cell density and soma size in the contralesional cortex, confocal stacks of 10 μm were taken (four sections per animals) with a step size of 1 μm using the FV1000 Olympus microscope (10× objective NA 0.4, zoom of 1.0 and 1024 × 1024). Fluorogold labeling was imaged using the 405 nm laser (with emission wavelength set at 610 nm), GFP with 488 nm laser and NeuN with 543 nm laser, in sequential scans.

*Spine density and morphology.* To measure spine density and morphology we relied on the intense and sparse labeling of GFP-M mice (essentially layer V neurons and fewer layer II/III neurons). In order to quantify the spine density/morphology of individual dendrites, we acquired z-stacks using the FV1000 Olympus microscope (60x oil objective NA 1.35, zoom of 3.5 and resolution of 800 × 800). We imaged three areas for each individual layer V neuronal dendrites and two areas for layer II/III neuronal dendrites (average length of 60 μm): proximal dendrite (at a distance of ~70–150 μm from soma), distal dendrite (at a distance of ~370–450 μm from soma, only for layer V neurons) and apical tuft (at a distance up to 75 μm below the S1 cortical surface). The images underwent deconvolution using the Huygens Essential software (16.05 Scientific Volume Imaging, NL) and Z projections of the deconvolved stacks were obtained using Neuron Studio.

**Data analysis.** All analysis were performed by an investigator blinded to the injury status and injury time points.

*Cortical thickness.* To calculate cortical thickness the distance between the bottom of Layer VI and top of Layer I, using the straight line tool in Fiji was made at three different coordinates in the medio-lateral direction (1 mm, 1.5 mm and 2 mm from midline). An average of those three measurements was then made.

*Cell density and cell size.* Images were processed and analyzed using the Definiens Developer XD™ software from Definiens, Version 2.7.0 which allowed a precise cell count. The cell counts were then divided by the area to obtain the cell density results.

*Spine density and morphology analysis.* Each deconvoluted stack containing one proximal, distal or apical tuft dendritic segment was opened in Neuronstudio (version 0.9.92) (Fig. 2.b) and the dendritic trunk was semi-automatically traced resulting in a series of green vertices superimposed on volume-rendered data. Thereafter, dendritic spines were detected and classified. This automatic quantification was manually controlled to remove falsely detected spines, add falsely undetected or reclassify misclassified spines. The resulting numbers of spines were divided by the length of the dendritic segment as calculated by the program.

*Analysis of synaptic input onto callosal dendrites.* Immunofluorescence staining was performed to visualize excitatory or inhibitory synaptic pairs together with RFP positive neuronal structures. Images were acquired on a Leica Malpighi TCS SP8 confocal microscope (63× oil objective, zoom of 3.5 and resolution of 1024 × 1024, z-stack of 0.3 μm). Quantification was carried as follows: the perimeter of the dendrites was outlined and measured. Then the number of excitatory or inhibitory synaptic pairs (in which both the pre-synaptic and post-synaptic markers co-localize) that co-localize with the outlined dendritic segments were counted. The density of synaptic pairs along the dendritic segments was then calculated.

*Lesion volume.* To characterize the lesion volume, we cut 50 μm coronal brain sections beginning from the first visibly "damaged" section to the last. Lesion volume was calculated according to Yu et al[71]. by outlying in 15–17 consecutive sections the area of the lesion 42 days following the TBI.

**In vivo 2 photon imaging of spine density in the contralesional cortex**
*Chronic cranial window implantation.* Before in vivo two-photon imaging, all animals were implanted with a chronic cranial window (CW), following a slightly modified protocol by Holtmaat et al. (2009)[72]. Mice were anaesthetized and fixed into a stereotactic frame. The skin above half of the skull was permanently removed with scissors. The remaining skin, surrounding the exposed skull was glued to the sides of the skull with Vetbond (3 M™ Vetbond™ tissue adhesive, 3 M United States). The CW (a circle of 4 mm of diameter) was positioned onto the somatosensory cortex between both coronal sutures and lambda and bregma, respectively. Then the bone was thinned until it was possible to remove the bone flap from the skull with forceps leaving the sagittal suture intact. A 4 mm diameter glass coverslip was

then positioned on top of the exposed and then glued and fixed onto to remaining skull, by applying another cyanoacrylate tissue adhesive (Histoacryl®, B.Braun) on the edges of the glass. Dental cement was then placed on the remaining skull, as well as at the edges of the glass coverslip, thereby securing the coverslip further. After the surgery, a resting period of at least 2–3 weeks before starting the imaging procedure was respected to minimize inflammation.

*Two-photon microscopy.* All animals were imaged using a two-photon microscope (Olympus FV1000 MPE), equipped with a Mai Tai DeepSee femto-second pulsed Ti:Sapphire laser (SpectraPhysics). In more detail, the apical dendritic tufts of GFP-labelled neurons of the somatosensory cortex were imaged. Before every imaging session animals were anaesthetized with an i.p. injection of MMF. Mice were placed under the two-photon microscope objective (Olympus XLPlan N 25X W MP). To visualize GFP-fluorescence, a green/red Olympus filter cube (FV10-MRVGR/XR; BA595-540, BA575) was used. The mice were imaged at a wavelength of 940 nm. A low-resolution overview stack was taken over regions of interest (resolution: 640 × 640 pixels; Zoom: 1; step 3 or 5 µm). This overview stack was later used to find back the same ROIs and dendrites each subsequent imaging session. In addition, the unique vascular arrangement, visible in superficial layers of the brain through the fluorescence lamp of the microscope, helped locate the same dendrites each following imaging period. For a detailed stack of individual dendrites and their spines, which were later used for spine analysis, a higher resolution image was acquired (resolution: 2048 × 2048 pixels; Zoom: 2; step 1 µm). In general, at least two detailed images with a varying number of analyzable dendrites (ranging from 3 to 20) were imaged per animal. Particular care was taken to keep a similar fluorescence intensity and laser power for each ROI, but also in between animals. Image acquisition was performed for baseline measurements and every 3 days after the induction of TBI. In total 2 baselines were acquired, baseline 1 (B1) and baseline 2 (B2), with 3 days in between each other. We imaged and followed 81 dendritic stretches from 4 animals (Fig. 1d) and a total of 10,191 spines throughout the experiment. The average length of the dendritic stretches quantified was 63.31 ± 1.54 µm.

*Analysis of in vivo spine dynamics and counting criteria.* Image analysis was performed only on the high resolution stacks with Fiji (ImageJ) software. Images were median filtered ("Despeckle"), and suitable dendrites for counting were determined. The length of each dendrite was traced and measured with Fiji at baseline. Dendritic spines were counted using the Fiji Cell Counter Plugin at each time point, by going through stacks manually and looking at each plane. In short, only structures clearly protruding laterally from the dendritic shaft, with a minimum protrusion length of 0.4 µm, were defined and counted as spines[72]. Structures, however, that fulfilled these criteria but coincided with another crossing dendrite and could not be distinctly attributed to the chosen dendrite were not included in the analysis. The total spine number, number of eliminated and formed spines, as well as the stable number of spines were determined. A spine was defined as eliminated, if it was not visible anymore in the next time point or as formed, if a new structure appeared, at the same location or where no spine had been counted before. Spines that were visible in successive time points at the same location were considered as stable. In addition, an elimination and formation rate was calculated by dividing the number of eliminated or formed spines at one time point by the number of total spines in the previous time point and multiplied by 100 (thus, rates were expressed as percentages). Spine density in turn was calculated relative to the length of the dendrite (number of spines/µm).

**In vivo 2 photon imaging of newly formed spine persistence index.** Imaging of the specific cell population of callosal and non callosal CST neurons (Fig.3) was carried out as described above. For detecting callosal neurons, we first injected the somatosensory cortex (coordinates from Bregma: rostrocaudal −1.5 mm, lateral 1.7 mm, depth 0.3 mm) with 0.5 µl of a retrograde AAV expressing GFP (Retro-AAV-GFP #37825 Addgene; Fig. 3a) and then implanted a CW 4 days later. To image non-callosal (control) neurons, we injected the dorsal column of cervical spinal cord with 0.5 µl of the same RetroAAV-GFP (Fig. 3d) and implanted the CW on the same day. Two to 3 weeks between the CW implantation and the first imaging session was given for all imaging groups in order to minimize possible inflammation. As with GFP-M animals 2 baseline measurements were done each 3 days apart before injury induction with CCI. Thereafter animals were imaged every 3 days and up to 42 days after TBI. Using this approach we could follow 23 apical dendrite stretches (8724 spines followed at each time points) for callosal neurons and 13 apical dendrite stretches (4376 spines followed at each time points) for CST neurons for dendritic lengths comprised between 62 and 88 µm

*Quantification of the persistence index.* To assess the stability of spines we determine the persistence index for both pre-existing spines (already present at the two baselines) and those that were newly formed after injury. Thus for pre-existing spines, single spines on dendrite stretches were followed at each imaging time-point and attributed to a binary code of either "1" when present or "0" when eliminated at every time-point individually. Then, the sum of binary code was divided by the number of all imaging time-points. For spines that were not present at baseline, but were newly formed after injury, the same persistence index was calculated[73].

**Neuronal connectivity**

*Viral tracing.* For labeling and analysis of neuronal cell connectivity using clearing we used the mono-synaptic rabies virus tracing technique. For this purpose, callosal axons on the ipsilesional side of the injury were first selectively infected with 0.5 µl of AAV expressing the cre recombinase ("AAV-cre" [AAV1.hSyn.Cre.W-PRE.hGH], titer: 5.0 e13 vg/ml supplied by the Penn Vector Core; dilution 1:6 in 1× PBS; coordinates from Bregma: rostrocaudal −1.5 mm, lateral 1.7 mm, depth 0.3 mm). For the acute time point groups (7 dpi) 0.5 µl of the helper virus AAV-TVA-G-eGFP ([AAV1-synP-DIO-sTpEpB-eGFP]; titer: 3.9×10e12 GC/ml supplied by Gene Therapy Vector Core, University of North Carolina; dilution 1:4 in 1× PBS) was injected on the same day, using the same coordinates as for AAV-cre but on the contralesional hemisphere. In contrast, chronic time point groups (42 dpi) were injected with AAV-TVA-G-eGFP, in the same way as described above, only 35 days after AAV-cre injection. In a final step, pre-synaptic partners of callosal neurons were visualized by injecting 1 µl of a rabies virus (SAD-ΔG-mcherry (EnvA); titer: 3 × 10^8 ffu/ml; dilution 1:1 in 1× PBS; coordinates from Bregma: rostrocaudal −1.5 mm, lateral 1.7 mm, depth 0.3 mm; kindly provided by K.K. Conzelmann[74]. Furthermore, for acute time points, TBI was induced on the same day as the rabies virus injection (2 weeks after the AAV-cre and AAV-TVA-G-eGFP injections), while animals in the chronic time point group were injured 2 weeks after AVV-cre- injection, 1 week before the AAV-TVA-G-eGFP and 35 days before the rabies virus injection. Control unlesioned animals were generated similarly, with one early and one late injection group.

*Tissue clearing.* Brains underwent staining and clearing using vDISCO protocol as described in Cai et al[42]. Briefly, after perfusing the animals with 1× PBS and 4% PFA, brains were collected and post-fixed for one night in 4% PFA and the next day were washed with 1× PBS.

Then whole brains were placed in 5 mL tubes (Eppendorf, 0030 119.401) and incubated in a permeabilization solution containing 1.5 vol/vol% goat serum (Gibco, 16210072), 0.5 vol/vol% Triton X-100 (AppliChem, A4975,1000), 0.5 mM of methyl-β –cyclodextrin (Sigma, 332615), 0.2% wt/vol% trans-1-acetyl-4-hydroxy-l-proline (Sigma-Aldrich, 441562) and 0.05 wt/vol% sodium azide (Sigma, 71290) in 1× PBS for 1 day at 37 °C in dark with gentle shaking. Then samples were incubated in 4.5 mL of a solution containing 1.5 vol/vol% goat serum, 0.5 vol/vol% Triton X-100, and 0.05 wt/vol% sodium azide in 1× PBS + 4uL of anti-RFP nanobody conjugated with Atto647N (Chromotek, rba647n-100, lot 81106002SAT2) for 14 days at 37 °C in dark with gentle shaking to stain the cells expressing mcherry. After 14 days, samples were washed with the same solution containing 1.5 vol/vol% goat serum, 0.5 vol/vol% Triton X-100, and 0.05 wt/vol% sodium azide in 1× PBS for four times every hour and finally with 1×PBS four times every hour. Both washing procedures were carried at room temperature in the dark and with gentle shaking. Samples were next cleared by incubating them in the following serial solutions for 2–3 h each step: 50 vol% THF, 70 vol% THF, 80 vol% THF in distilled water, 100 vol% THF overnight and again 1 h in 100 vol% THF, followed by 2 h in dichloromethane and finally in BABB until samples turned completely transparent after 5–6 h. All the clearing steps were performed with gentle shaking, at room temperature and by protecting the samples from light.

*Light-sheet microscopy imaging and whole brain analysis.* Cleared brains were imaged as tiling scans (10% overlap) with the light-sheet microscope Ultramicroscope II (LaVision BioTec) using a x2 objective Olympus MVPLAPO2XC/0.5 NA (WD = 6 mm) coupled to an Olympus MVX10 zoom body that was kept at zoom magnification ×2.5 using the filter set ex 640/40 nm, em 690/50 nm. Stitching of tile scans was performed by using Fiji's (ImageJ, v.1.52 h, https://fiji.sc/) stitching plugin[75]. Stitched images were saved in tiff format. Fiji's TrakEM2 plugin and Imglib2 library were used to correct acquisition shifting[76]. Scans were pre-processed in Fiji software for background equalization via pseudo-flat-field correction function from Bio-Voxxel toolbox, for background removal (to remove particles bigger than cells) via median option from the same toolbox, for noise reduction via two-dimensional medial filter and for signal amplification via the unsharpen mask. Cells were visualized as heatmaps after Clearmap software analysis as described by Renier et al[77]. TBI and control heatmaps colored with two different colors were overlayed over the brain atlas provided by Clearmap and merged in Fiji and displayed as image sequence in Fig. 5b and in the Supplementary movie.

*Tissue processing and immunohistochemistry.* For circuit connectivity analysis, mice were transcardially perfused either 7 or 42 days after TBI, as described above. Control mice were sacrificed on the same day as TBI-mice. After the 24 h period of post-fixation in 4% PFA, brains were microdissected and kept in eppendorf receptacles with 30% sucrose until sinking to the bottom. They were sectioned coronally with a cryostat (Leica CM1850) at a thickness of 60 µm. Next, a GFP-amplification-protocol combined with a NeuroTrace (NT) staining was performed on free-floating sections. To this end, sections were first washed three times with 1×PBS à 10 min. In the final washing step, 1 × PBS was replaced by blocking buffer (1 × PBS with 0.5% Triton and 10% goat serum) and sections were incubated in it for 1 h at room temperature. After removing the blocking buffer, primary antibody solution (1× PBS containing 0.1% Triton, 1% goat serum and 1:500 chicken anti-GFP Ab (Abcam)) was added. Overnight incubation at 4 °C ensued. The following

day, sections were washed three times for 10 min with 1× PBS/0.1% Triton and later incubated with the secondary antibody solution (1× PBS −0.1% Triton, 1% goat serum and 1:500, Goat Anti-Chicken Alexa Fluor® 488 Ab (Abcam); 1:500 NT435 (Invitrogen)) overnight at 4 °C. After a last washing step, three times à 10 min with 1xPBS, sections were mounted on gelatine subbed slides with Vecta-Shield (Vector Laboratories) and finally coverslipped as described above.

*Imaging of cleared tissue.* Images of brain sections of 2 mm thickness were acquired on a light-sheet microscope (UM II LaVision BioTec-UltraMicroscope II), with a z-step size of 3 μm, using a 4× objective (0.28 NA, Olympus XLFLURO4x), with a lens provided with a 10 mm working distance dipping cap. Images were processed with Imaris (Bitplane1) in order to generate 3D-projections, while HeatMap Histograms were created using Fiji.

*Imaging.* Starter cells were evaluated as follows: A total of six sections per brain, beginning from the section with the first detectable starter cells to the section with the last detectable starter cells, were scanned using a Leica SP8 confocal microscope (magnification: ×20; Zoom 1; step; 1 μm; resolution: 512 ×512 pixels). Presynaptic cells were evaluated as follows: Here, 20 sections per brain were selected according to the Allen Mouse Brain Atlas from ~1.15 to ~2.85 mm relative to Bregma[78]. All sections were imaged consecutively with a Leica DM4 B microscope (Leica Microsystems), using the MBF Stereo Investigator 2017 software and a 2.5X objective when cells were sparse. In cases where cell labelling was too dense we used a 10X objective. All 2.5 and ×10 magnification images were stitched with the Fiji Pairwise Stitching Plugin. The 20X image stacks were further processed ImageJ (Fiji).

*Quantifications.* Starter cells were manually quantified using the Fiji Cell Counter Plugin in all imaged sections and interpolated to the total number of sections evaluated in the whole brain. For quantification, each imaged section was sub-divided into different brain areas, as defined by the Allen Mouse Brain Atlas. In total we chose 20 different brain regions, ranging from cortical to subcortical regions as well as white matter tract structures. Within those, pre-synaptic cells (m-Cherry positive) were manually counted with the Fiji Cell Counter Plugin. For areas in which cell aggregation was too dense and manual counting was not accurate enough we employed Fiji's Particle Analyzer Plugin for automatic quantification, after image processing for background subtraction. We then made a ratio between the total number of pre-synaptic neurons in that specific area to the total number of starter cells in the whole brain. This ratio was termed connectivity ratio, as previously described[43]. We then compared the connectivity ratios of acute and chronic injured animals with their respective controls (acute and controls generated with the respective virus injections timelines as for injured animals).

## Calcium imaging experiments

*Viral injections and window implantation.* For in vivo calcium imaging, callosal neurons were selectively traced and labelled using viral vectors. To do so we injected mice with a retrograde adeno-associated virus (retro AAV-CAG-Td -Tomatoe, Addgene, 59462-AAVrg; Titer≥7 × 10^{12} vg/mL; dilution 1:3 with 1× PBS) as described above. AAV1.hSyn-GCaMP6mWPRE.SV40 (Addgene 100838, Titer ≥10^{13}vg/mL) was injected in the contralesional site at the following coordinates (rostro-caudal to Bregma: −1.8; lateral to Bregma 2.0; 0.3 mm depth) and a CW (diameter 4 mm) was implanted above as described above.

*Two-photon imaging in anesthetized mice.* In vivo two-photon imaging was performed on a resonant scanning two-photon microscope (Hyperscope, Scientifica, equipped with an 8 kHz resonant scanner) and a 16× water-immersion objective (Nikon), yielding frame rates of 30 Hz at a resolution of 512 × 512 pixels. We recorded 18,000 frames (10 min) within cortical layer II/III (cortical depths of 140–310 μm), covering a field of view (FOV) of 230 × 230 μm². Light source was a Ti:Sapphire laser with a DeepSee pre-chirp unit (Spectra Physics MaiTai eHP). GCaMP6m and tdTomato were co-excited at 930 nm, with a laser power not exceeding 40 mW (typically 10–40 mW). During imaging, mice were anesthetized with 1.0–1.5% isoflurane in pure O₂ at a flow rate of ~0.5 l/min, to maintain a respiratory rate in the range of 110–140 breaths per minute. Body temperature was maintained at 37 degrees using a physiological monitoring system (Harvard Apparatus). In total 663 non-callosal and 139 callosal neurons were analyzed within 13 experiments in 6 mice.

*Image processing and data analysis.* All image analyses were performed using custom-written routines in Matlab (R2018b)[79]. In brief, full frame images were corrected for potential x and y brain displacement occurred during the in vivo recording period, and regions of interests (ROIs) were manually selected based on the maximum and mean projection of all frames. Those ROIs were readjusted, if necessary, over the different recorded frames following our time course. Fluorescence signals of all pixels within a selected ROI were averaged, the intensity traces were low pass filtered at 10 Hz. Contamination from neuropil signals was accounted for, as described using the following equation: $F_{ROI\_comp} = F_{ROI} - 0.7 \times F_{neuropil} + 0.7 \times$ median $(F_{neuropil})$

$F_{ROI\_comp}$ stands for neuropil-compensated fluorescence of the ROI, $F_{ROI}$ and $F_{neuropil}$ represent the initial fluorescence signal of the ROI and the signal from the neuropil, respectively. A neuron was defined as 'active' if it displayed at least one prominent calcium transient over 20 frames (corresponding to ~0.7 s).

**Statistical analysis**. All results are given as mean ± standard error of the mean (SEM), unless otherwise stated. Statistical analysis, as well as graphs illustrating data, was carried out with GraphPad Prism 7.01 for Windows (GraphPad Software). Ladder rung behavior data were analyzed with a Friedman test followed by post-hoc test. In vivo two-photon imaging data were analyzed using one-way and two-ways RM ANOVA followed by post-hoc test. In experiments without data of a "repeated-measures-type", we performed an ordinary one-way ANOVA and post-hoc test. In addition, two-tailed t-test were carried out for connectivity analysis and in vivo functional imaging experiments (unless otherwise stated). To compare cumulative distributions we used the Kolmogorov–Smirnov (KS) test. When data were unparametric we used Mann–Whitney test to compare two groups. Statistical significance levels are indicated as follows: $*p < 0.05$; $**p < 0.01$; $***p < 0.001$.

**Reporting summary**. Further information on research design is available in the Nature Research Reporting Summary linked to this article.

## Data availability
Source data are provided with this paper and have also been deposited in the Zenodo database under https://doi.org/10.5281/zenodo.6364730 including supplementary data. Raw data are available from the corresponding author upon reasonable request.

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

## Acknowledgements

We thank B. Fiedler and L. Schödel for excellent technical assistance as well as D.Matzek and B. Stahr for animal husbandry. F.M.B. is supported by the Deutsche For-schungsgemeinschaft (DFG, SFB 870 Project ID 118803580). F.M.B. is supported by the Wings for Life Foundation, the International Foundation for Research in Paraplegia (IRP) and the DFG, TRR274 (Project ID 408885537). F.M.B. and S.L. are supported by the Munich Center for Systems Neurology (DFG, SyNergy; EXC 2145/ID 390857198). S.L. is supported by the Emmy Noether program (DFG). V.V.S. is supported by the Humboldt foundation and J.F. is supported by the Wings for Life Foundation. The authors declare no competing financial interests. We thank the Core Facility Bioimaging at the Biomedical Center (BMC) for excellent imaging support.

## Author contributions

F.M.B. designed the experiments. L.E., A.C., M.C., W.Y.V.K. contributed all surgical procedures. L.E., A.C., M.C., W.Y.V.K., S.W., V.V.S., J.F., M.M. collected and analyzed data. A.G., P.B., and K.K.C. contributed rabies viruses. R.C., A.G., A.E. contributed clearing experiments. M.C., W.I.V.K., and S.L. contributed calcium imaging and analysis. I.W., M.K., and D.M. analyzed cell density in the cortex and performed paraffin-embedded immunohistochemistry. L.E., A.C., S.L., and F.M.B. wrote the paper. All authors approved the final version of the paper.

## Funding

## Competing interests

The authors declare no competing interests.
