## [Peer Review File · Nature Communications]

Reviewers' Comments:

Reviewer #1:

Remarks to the Author:

The manuscript describes changes to callosal projecting neurons in the contralesional hemisphere of a traumatically-injured mouse brain. The authors have used a vast array of complementary techniques to demonstrate structural and functional alterations in these neurons and report that these changes are exclusive to the callosal neurons rather than the non-callosal neurons.

1. The rationale for the study is that mice (and humans) demonstrate marked recovery following TBI. The current study would have been significantly strengthened if the authors had demonstrated that motor and sensorimotor function in these animals had changed between the early (7d) and late (42d) time points.
2. One of the issues with such an exhaustive analysis is that the reader can get lost in the weeds. It might of service to the readers if the authors were to add a table that summarizes the changes.
3. As this reviewer understands the initial analysis (figure 2) was performed on layer V neurons. It appears that the data from the in vivo calcium imaging and from the retrograde tracing was more focused on layers 2-4. If this is not the case, then the authors should provide a more detailed description of cortical layers that were assessed in all outcomes. If the layers are different, the authors should discuss the potential differences across outcomes.
4. Supplementary figures 3 and 4 provide important information that allows the reader to better absorb the importance of the studies and it is this reviewers recommendation that these figures be included in the manuscript and not relegated to supplemental data.
5. It would be of benefit if the authors could comment on the correlation of the lesion size in the ipsilesional hemisphere to the changes in the structure and function of the neurons in the homotypic cortex of the contralesional hemisphere.
6. Some details regarding the injury model should be provided
 - (a) was the impactor tip hemispheric or flat-edged?
 - (b) what were the dimensions of the rectangular craniotomy?
 - (c) how were the animals classified as being moderately injured? Apnea and righting reflex recovery times should be reported. If multiple groups of animals were used for the various outcomes, these data would be useful to appreciate the reproducibility across these groups.
 - (d) was the analysis performed in an unbiased manner? How?
 - (e) the source and sex of the animals must be reported. If only one sex was used, a justification must be provided.
7. With TBI in mice, the blood-brain barrier is breached leading to entry of blood and blood cells into the brain. Using a monoclonal antibody for immunofluorescence poses problems with respect to non-specific binding of the secondary antibody. The authors should confirm that this was appropriately controlled for.

Reviewer #2:

Remarks to the Author:

This is a very thorough and important study looking at the neural population and circuitry implicated in post TBI plasticity. It is well written and the results showing the contralateral neurons are specifically modulated after injury is important to design future strategies for recovery. The combination of anatomical tracing and functional calcium imaging is impressive.

This study also builds on previous works that demonstrated similar functional findings, that callosal neurons and synapses is significantly impacted by TBI. I suggest including this literature, specifically: PMID 24050267 and 26440604.

In addition, I suggest addressing the following:

Moderate TBI- what is the expected anatomical damage and how was it assessed. It would be beneficial to include a figure with actual anatomical images of the lesion area.

The lesion size was determined to be over 7 mm³. The entire mouse brain is 11 mm in length. 7 mm seems like a very significant lesion size... is this injury considered 'moderate', and to what extent are we expecting a recovery after an injury with this magnitude?

Spine density- what could be the scientific reason for significance reduction in spines before the injury? Figure 1 D shows that the only time point where there is actually elimination is spine density occurs before injury (B2). That might be a typo?

Reviewer #3:

Remarks to the Author:

This is an elegant set of experiments to characterize patterns of synaptic remodeling, re-connectivity and neural activity changes in contralesional homotopic cortex over time after a cortical impact injury. This was achieved using clever combinations of repeated cranial window imaging and viral tracing approaches in the mouse controlled cortical impact model (CCI).

The findings are significant for advancing basic understanding of regenerative reactions to brain injury. The specificity of the effects in contra hemisphere to the synaptic connections most directly affected by the injury is itself a finding of significance for understanding spine remodeling responses.

Issues:

The significance of the findings for illuminating the mechanisms contributing to behavioral recovery is not clear. It is oversold in the presentation. There is very good evidence that changes in contralesional cortex can sometimes contribute to recovery, but there is also evidence that it does not always do so (e.g., Dancause et al., 2015, *Prog Brain Res*; Bradnam et al., 2012 *Cerebral Cortex*; Shanina et al., 2006 *Neurosci*), and that neuronal structural remodeling responses in contra cortex need not mediate recovery (e.g., Jones, 2017, *Nat Rev Neurosci*). In this context, and given that the present studies do not establish a relationship of the observed changes with behavioral recovery, the relevance of the findings to mechanisms of recovery is unclear.

The characterization of TBI effects on spontaneous activity of callosal and non callosal cells is interesting but a link with the structural changes described is also not fully resolvable. The studies do not establish definitive links between activity and structural patterns of change.

Minor:

The analyses of spine turnover responses treated dendritic segments (stretches) as individuals (n = 81) rather than segments within animals (n=4). This is understandable for this approach, but it would be comforting to know whether there is a considerable amount of inter-animal variability in the patterns observed. Does the mean per stretch follow a similar pattern? One possible way to capture this would be to color code dots in Fig. 1c per animal, if it does not add too much confusion.

The authors might consider reporting on rates of new spine survival. The continued high rate of spine elimination over 18 days suggests that many of them were being eliminated, but presumably some would persist to contribute to the ultimate normalization of spine density.

The definition of apical tufts for the histological analyses of spine density was based on distance from soma (p. 6). Given variation in the depth of the soma in layer V, this could be suboptimal for maximizing overlap with the population of dendrites samples *in vivo*. This might be resolved simply by reporting the depth relative to the cortical surface of the sampled apical tufts.

Point to point response

Reviewer #1 (Remarks to the Author):

The manuscript describes changes to callosal projecting neurons in the contralesional hemisphere of a traumatically-injured mouse brain. The authors have used a vast array of complementary techniques to demonstrate structural and functional alterations in these neurons and report that these changes are exclusive to the callosal neurons rather than the non-callosal neurons.

We thank the reviewer for appreciating our work and our experimental approach.

1. The rationale for the study is that mice (and humans) demonstrate marked recovery following TBI. The current study would have been significantly strengthened if the authors had demonstrated that motor and sensorimotor function in these animals had changed between the early (7d) and late (42d) time points.

We agree with the reviewer that showing recovery of sensorimotor function would further strengthen the manuscript. We have therefore conducted the ladder rung test that evaluates sensorimotor function in mice at early and late timepoints following TBI. Our results demonstrate that indeed deficits of sensorimotor function persist for the first 7 days after TBI and subsequently recover in parallel to the anatomical remodeling process that is likely to include callosal but also corticospinal and intraspinal pathways. This new experiment has now been added to Figure 1 of the revised manuscript e.g. (Fig. 1b) as well as to the revised Material and Methods (lines 476-486) and the Result (lines 135-144) section of the manuscript.

2. One of the issues with such an exhaustive analysis is that the reader can get lost in the weeds. It might of service to the readers if the authors were to add a table that summarizes the changes.

This is an excellent idea that we now have implemented to summarize the changes observed during this study. The table is now added at the end of the paper as a new Table 1.

Table1:

Manipulation	Acute time points	Later time points	Figure panel
Sensorimotor function	Decreased	Normalized	1b
Spine density in the contralesional cortex	Decreased	–	1e
Spine density of callosal neurons	Decreased	Normalized	2c & d
Spine density of non callosal neurons	Not changed	Not changed	2e & f
Excitatory input onto callosal neurons	Decreased	–	2g & h
Inhibitory input onto callosal neurons	No changed	–	2g& h
Spine morphology of callosal neurons	Altered	Normalized	3b & c
Spine morphology of non callosal neurons	Not changed	Not changed	3d & e

Persistence of new spines on callosal neurons	–	Higher in callosal neurons compared to control neurons	4g
Connectivity ratios of callosal neurons	Decreased	Normalized	5
Activity patterns callosal neurons	Decreased	Normalized	6c
Activity patterns non callosal neurons	Decreased	Normalized	6c

3. As this reviewer understands the initial analysis (figure 2) was performed on layer V neurons. It appears that the data from the in vivo calcium imaging and from the retrograde tracing was more focused on layers 2-4. If this is not the case, then the authors should provide a more detailed description of cortical layers that were assessed in all outcomes. If the layers are different, the authors should discuss the potential differences across outcomes.

The reviewer is correct in pointing out that the initial analysis was performed in layer V of the cortex in GFP-M mice as most of the labelled neurons in this mouse line are layer V neurons. However as GFP-M mice also contain some labelled layer II-III neurons, we have now analyzed the spine density and morphology in those layer II-III neurons in order to further strengthen the link between our fixed tissue and in vivo analyses. Indeed these new analyses show that layer II-III callosal neurons (but not the non callosal neurons in these layers) demonstrate a loss of spines and alterations in spine morphology at acute time points following TBI that subsequently recover recapitulating the changes observed in layer V callosal neurons. We have now added these data to the revised Figure 2 (Fig. 2d,e) and present them together with those derived from Layer V callosal neurons. We have further revised our Results (lines 177-194) and Methods (lines 556-560) sections to include this new analysis.

4. Supplementary figures 3 and 4 provide important information that allows the reader to better absorb the importance of the studies and it is this reviewers recommendation that these figures be included in the manuscript and not relegated to supplemental data.

We agree with the reviewer that the analyses of spine morphology in additional dendritic segments of callosal neurons and in non callosal neurons is important and should be added to the main manuscript. As we have now in addition analyzed the spine density and morphology for layer II/III neurons (see response above), we decided that it would be too much for one figure. We have therefore separated the spine density data (new Fig.2) from the spine morphology data (new Fig.3) in two separate figures and are now presenting the analyses derived from all dendritic segments of callosal and non callosal neurons in those figures to let the reader appreciate the selectivity of the response pattern. Therefore, the two supplementary figures are now removed from the paper. We have adapted the figure legends accordingly.

5. It would be of benefit if the authors could comment on the correlation of the lesion size in the ipsilesional hemisphere to the changes in the structure and function of the neurons in the homotypic cortex of the contralesional hemisphere.

We are now providing the lesion size for a subset of animals as well as a representative confocal image in the revised Supplementary Fig.1. These results (together with the analyses of apnea duration and righting reflex (see response to point 6, below) are in line with a classification of our

lesions as “moderate” lesions, which in the past have been shown to favor remodeling¹⁻³. Such remodeling indeed appears to be less prevalent in either smaller or larger lesions. We now discuss this possible relation between the lesion extent and the recovery process in the contralesional homotypic area in the revised discussion section (lines 388-394). It was also added to the Material and Method section (lines 588-590).

6. Some details regarding the injury model should be provided

(a) was the impactor tip hemispheric or flat-edged?

(b) what were the dimensions of the rectangular craniotomy?

(c) how were the animals classified as being moderately injured? Apnea and righting reflex recovery times should be reported. If multiple groups of animals were used for the various outcomes, these data would be useful to appreciate the reproducibility across these groups.

(d) was the analysis performed in an unbiased manner? How?

(e) the source and sex of the animals must be reported. If only one sex was used, a justification must be provided.

As suggested by the reviewer we are now providing more information about our injury paradigm including related to the the size of the craniotomy, the tip shape and the degree of injury in the revised Methods section (lines 465-468). We are also now providing the information about righting reflex and apnea (Supplementary Fig. 1) to show that our TBI settings induce a moderate injury⁴⁻⁶. All analysis were performed by an observer blinded to the injury status and the injury time points. Sections compared in the same analysis were stained in parallel and imaging was performed using standardized protocols (as described in the Methods section). In those experiments performed with C57Bl6 mice we used only female mice as these are less likely to fight during the recovery phase. In those experiments requiring transgenic labelling, we needed to use both male and female mice due to scarcity of transgenic mice. These were randomly distributed to the experimental groups. We now provide these information in the Material and Method section (lines 454, 464,467, 470-472 & 565).

7. With TBI in mice, the blood-brain barrier is breached leading to entry of blood and blood cells into the brain. Using a monoclonal antibody for immunofluorescence poses problems with respect to non-specific binding of the secondary antibody. The authors should confirm that this was appropriately controlled for.

We used a mouse monoclonal antibody (NeuN) antibody for staining of neuronal cell bodies following injury. As the reviewer correctly points out we controlled for non-specific binding of the secondary (anti-mouse IgG) antibody by always including sections stained with the secondary antibody alone in our immunohistochemical analyses performed in control and injured mice.

Reviewer #2 (Remarks to the Author):

This is a very thorough and important study looking at the neural population and circuitry implicated in post TBI plasticity. It is well written and the results showing the contralateral neurons are specifically modulated after injury is important to design future strategies for recovery. The combination of anatomical tracing and functional calcium imaging is impressive.

This study also builds on previous works that demonstrated similar functional findings, that callosal neurons and synapses is significantly impacted by TBI. I suggest including this literature, specifically: PMID 24050267 and 26440604.

We thank the reviewer for the overall positive assessment of our work. We have added those important publications to the revised manuscript and now refer to these studies in the introduction and discussion (lines 82, 366 & 370).

In addition, I suggest addressing the following:

Moderate TBI- what is the expected anatomical damage and how was it assessed. It would be beneficial to include a figure with actual anatomical images of the lesion area.

We thank the reviewer for this question that was also raised by Reviewer 1. We have now conducted an analysis of the lesion volumes of our animals and provide the results of this quantification together with a representative confocal image of the associated injury. These new data sets are presented in a new Supplementary Figure 1 and are discussed in the revised discussion (lines 3889-394) and added to the Methods section (lines 470-472).

The lesion size was determined to be over 7 mm³. The entire mouse brain is 11 mm in length. 7 mm seems like a very significant lesion size... is this injury considered 'moderate', and to what extent are we expecting a recovery after an injury with this magnitude?

The lesion volume of 7 mm³ has to be considered in relation to the entire volume of the adult mouse brain which is estimated to be 508.91±23.42 mm³⁷. The cortex of adult C57Bl6 mice is estimated to be 169.61±8.88 mm³⁷, indicating that our lesion affects a bit more than 4% of the cortical volume. To further justify the classification of our TBI lesions as "moderate" we have now analyzed the apnea and righting reflex after injury. Taken together the measured lesion size and as well as the observed reflex responses in our model are consistent with a moderate injury⁴⁻⁶. As we agree with the reviewer that these information are important in the context of our paper, we have now added them to the paper (Supplementary Fig.1 and discussion (lines 388-394) and Material and Methods (lines 470-472 and 588-590). Furthermore to ensure that mice can indeed successfully recover from such an injury, we have now conducted behavioral studies to quantify the sensorimotor recovery in our model (see also Response to Reviewer 1, Point 1). We are now presenting the behavioral recovery following TBI in the new Figure 1 (Fig.1b).

Spine density- what could be the scientific reason for significance reduction in spines before the injury? Figure 1 D shows that the only time point where there is actually elimination is spine density occurs before injury (B2). That might be a typo?

This might have been a misunderstanding as the spine loss is not between B1 and B2 (the two baseline timepoints before injury) but between B2 and "3" (which refers to the first imaging post injury, at 3 days after the TBI) as shown in the original panel 1D (now panel 1F).

Reviewer #3 (Remarks to the Author):

This is an elegant set of experiments to characterize patterns of synaptic remodeling, re-connectivity and neural activity changes in contralesional homotopic cortex over time after a cortical impact injury. This was achieved using clever combinations of repeated cranial window imaging and viral tracing approaches in the mouse controlled cortical impact model (CCI).

The findings are significant for advancing basic understanding of regenerative reactions to brain injury. The specificity of the effects in contra hemisphere to the synaptic connections most directly affected by the injury is itself a finding of significance for understanding spine remodeling responses.

We thank the reviewer for her/his overall positive assessment of our work.

Issues:

The significance of the findings for illuminating the mechanisms contributing to behavioral recovery is not clear. It is oversold in the presentation. There is very good evidence that changes in contralesional cortex can sometimes contribute to recovery, but there is also evidence that it does not always do so (e.g., Dancause et al., 2015, *Prog Brain Res*; Bradnam et al., 2012 *Cerebral Cortex*; Shanina et al., 2006 *Neurosci*), and that neuronal structural remodeling responses in contra cortex need not mediate recovery (e.g., Jones, 2017, *Nat Rev Neurosci*). In this context, and given that the present studies do not establish a relationship of the observed changes with behavioral recovery, the relevance of the findings to mechanisms of recovery is unclear

We agree with the reviewer that the contribution of adaptive changes in the contralesional cortex to recovery of function are detected in some but not in all experimental settings. To better reflect the state of the literature we have therefore revised our introduction and extended the discussion of this aspect and included the additional citations suggested by the reviewer to better reflect the state of the research on this topic. We have also slightly changed our title and replaced " Adaptive plasticity of callosal neurons in the adult contralesional cortex following traumatic brain injury" by "Selective plasticity of callosal neurons in the adult contralesional cortex following traumatic brain injury".

Furthermore (and also following the suggestions by the other reviewers) we have now performed behavioral testing to demonstrate that the traumatic brain injury in our model is indeed followed by functional recovery that parallels the anatomical remodeling process (new Fig. 1b). However we acknowledge that we cannot formally demonstrate that the contralesional remodeling that we present here is directly responsible for the observed recovery of sensorimotor function (and it is in fact likely that similar remodeling processes in other CNS locations including of cortico-spinal and intraspinal connections contribute as well). We therefore make sure not to overstate the link that could exist between those experimentally separate findings. We believe that our discussion of the contribution of the contralateral cortex to functional recovery is now better balanced. The changes can be found in the abstract (lines 50-51 & 57-60), introduction (lines 84-90 and lines 98-102) and discussion (lines 353-356 and lines 388-394).

The characterization of TBI effects on spontaneous activity of callosal and non callosal cells is interesting but a link with the structural changes described is also not fully resolvable. The studies do not establish definitive links between activity and structural patterns of change.

We agree with the reviewer that it is challenging to directly link structural changes of neurons to activity patterns. To better address the relation between the structural and functional changes to callosal neurons after injury, we have now performed an additional set of experiments characterizing the changes to excitatory and inhibitory inputs of callosal neurons acutely following TBI. For this purpose we performed immunohistochemical analyses and identified excitatory and inhibitory synapses as close contacts between a pre-synaptic protein (Vglut1 for the excitatory pre-synaptic partner or GAD67 for the inhibitory pre-synaptic partner) and a post-synaptic protein (Homer for the excitatory post-synaptic partner or Gephyrin for the post-synaptic inhibitory partner) on thin paraffin-embedded sections and quantified the number of excitatory and inhibitory inputs onto the dendrites of retrogradely labeled callosal neurons before and acutely following TBI. Here we find that there is a marked decrease of the number of excitatory synapses onto callosal neurons in the contralesional cortex at 7d after TBI. This loss of excitatory inputs is consistent with the decrease in activity we observe in the same neurons at acute timepoints following injury^{8,9}. We have now added these data to Figure 2 (**Fig. 2g,h**) and accordingly extended the discussion of the relation between structural and functional changes of transcallosal neurons after injury (lines 418-423). We therefore completed the Material and Method section with this new experiment (lines 502-505, lines 514-518 and lines 528-541) and the Results section (lines 202-208).

Minor:

The analyses of spine turnover responses treated dendritic segments (stretches) as individuals (n = 81) rather than segments within animals (n=4). This is understandable for this approach, but it would be comforting to know whether there is a considerable amount of inter-animal variability in the patterns observed. Does the mean per stretch follow a similar pattern? One possible way to capture this would be to color code dots in Fig. 1c per animal, if it does not add too much confusion.

We thank the reviewer for this suggestion and have now color-coded the graphs as suggested to better show inter- and intra-animal variations in the new Fig. 1e.

The authors might consider reporting on rates of new spine survival. The continued high rate of spine elimination over 18 days suggests that many of them were being eliminated, but presumably some would persist to contribute to the ultimate normalization of spine density.

We have followed the reviewer's suggestion and performed a new experiment to follow the fate of pre-existing spines (already present before the brain injury) as well as newly formed spines after TBI. To do so, we retrogradely labeled callosal neurons and non callosal neurons (here we choose CST neurons as these are located in the same area as callosal neurons but do not project to the lesioned hemisphere) with a retroAAV expressing eGFP and analyzed spine persistence of pre-existing as well as newly-formed spines by repetitive in vivo imaging before and for 42 days after TBI. First this analysis confirms that – as expected – pre-existing spines are more stable than newly formed spines. Notably however, when we analyzed the data further and separated newly formed spines that form on callosal dendrites from those that form on non callosal (CST) dendrites, we observed that newly-formed spines on callosal neurons persist substantially longer than newly formed spines on non-callosal neurons. This finding provides a mechanistic explanation for the recovery of spine densities on callosal neurons over time and underlines the functional significance of the associated rewiring of the callosal input circuitry. We have included these exciting new findings as a new Figure 4 and also

added them to the revised Methods (lines 646-667) and Results sections (lines 236-255) and discuss its implications in the revised discussion (lines 376-384).

The definition of apical tufts for the histological analyses of spine density was based on distance from soma (p. 6). Given variation in the depth of the soma in layer V, this could be suboptimal for maximizing overlap with the population of dendrites samples in vivo. This might be resolved simply by reporting the depth relative to the cortical surface of the sampled apical tufts.

We agree with this assessment. We are now reporting the mean depth from the cortical surface for the sampled apical tufts. In essence, we included all tufts up to 70 mm below the S1 cortical surface (now mentioned lines 556-560) of the revised Methods section).

References

1. Jones, T. A. Motor compensation and its effects on neural reorganization after stroke. *Nat. Rev. Neurosci.* **18**, 267–280 (2017).
2. Bradnam, L. V., Stinear, C. M., Barber, P. A. & Byblow, W. D. Contralesional hemisphere control of the proximal paretic upper limb following stroke. *Cereb. Cortex* **22**, 2662–2671 (2012).
3. Morecraft, R. J. *et al.* Frontal and frontoparietal injury differentially affect the ipsilateral corticospinal projection from the nonlesioned hemisphere in monkey (*Macaca mulatta*). *J. Comp. Neurol.* **524**, 380–407 (2016).
4. Pernici, C. D. *et al.* Longitudinal optical imaging technique to visualize progressive axonal damage after brain injury in mice reveals responses to different minocycline treatments. *Sci. Rep.* **10**, (2020).
5. Rowe, R. K. *et al.* Aging with Traumatic Brain Injury: Effects of Age at Injury on Behavioral Outcome following Diffuse Brain Injury in Rats. *Dev. Neurosci.* **38**, 195–205 (2016).
6. Harrison, J. L. *et al.* Resolvins AT-D1 and E1 differentially impact functional outcome, post-traumatic sleep, and microglial activation following diffuse brain injury in the mouse. *Brain. Behav. Immun.* **47**, 131–140 (2015).
7. Badea A, Ali-Sharief AA, Johnson GA. Morphometric analysis of the C57BL/6J mouse brain. *Neuroimage.* 2007;37(3):683-693. doi:10.1016/j.neuroimage.2007.05.046
8. Rose, T., Jaepel, J., Hübener, M. & Bonhoeffer, T. Cell-specific restoration of stimulus preference after monocular deprivation in the visual cortex. *Science* **352**, 1319–1322 (2016).
9. Jafari, M. *et al.* Phagocyte-mediated synapse removal in cortical neuroinflammation is promoted by local calcium accumulation. *24*, 355–367 (2021).

Reviewers' Comments:

Reviewer #2:

Remarks to the Author:

Authors had responded thoughtfully to the reviews.

Reviewer #3:

Remarks to the Author:

The authors have thoroughly addressed my prior concerns. The new analysis of excitatory vs. inhibitory synapses (Fig2 G & F) is impressive. The addition of new spine survival analysis (Fig. 4) further strengthens the study by more firmly linking the spine turnover response to spine density recovery. Color coding of individual mice in Fig. 1e is a nice touch that ensures confidence in these results. Other revisions have added appropriate nuance regarding the role of the contralesional hemisphere in recovery. Remaining issues are minor.

Minor:

Re new data on new spine persistence: Was the sample of new spines in control neurons (corticospinal tract neurons) large enough for sensitive analysis of new spine persistence in this population? The authors might address this simply by reporting the numbers of new spines/dendritic stretch per each of the neuron populations.

The time lapse series in Fig. 4C & F could be more helpful if newly formed spines that persisted vs. disappeared were indicated.

Re connectivity results –It would be helpful to report standard error with the mean connectivity ratios shown in Fig. 5's tables. Also please clarify whether the n=6-8 per group reported in Fig. 5's legend is per each time point (vs. pooled across them).

Reviewer #4:

Remarks to the Author:

The authors have been very responsive to the previous critique of their manuscript and have adequately addressed all substantive issues.

Two minor comments:

1. As a point of clarification regarding the statistical analyses, please denote the comparison indicated by the significance symbols in each figure. For example, in Figure 1, the meaning of # as compared to * is not defined in panel B and the reference group for these comparisons is not provided. In Figure 1e, are comparisons (*) relative to B1 or B2? In Figure 1f, does * denote a change from baseline (within a group) or a difference at a given time between 'eliminated' and 'formed' groups? Please check all figures for similar issues.
2. Please include the time point at which lesion volume was assessed.

Point to point response

Reviewer #3

Re new data on new spine persistence: Was the sample of new spines in control neurons (corticospinal tract neurons) large enough for sensitive analysis of new spine persistence in this population? The authors might address this simply by reporting the numbers of new spines/dendritic stretch per each of the neuron populations.

For figure 4g (top), the number of spines analyzed is the following: we analyzed 377 pre-existing spines and 187 newly formed spines to generate the data so a total of 564 spines. These numbers are now added to the figure legend. For figure 4g (bottom) we analyzed newly-formed spines: 55 control CST spines and 134 callosal spines. This has now been added to the manuscript lines 1146-1148 (figure 4g legend).

The time lapse series in Fig. 4C & F could be more helpful if newly formed spines that persisted vs. disappeared were indicated.

Thanks for the suggestion. We have now done so and added significances of the symbols in the figure legend lines 1142-1145 (figure 4f legend).

Re connectivity results –It would be helpful to report standard error with the mean connectivity ratios shown in Fig. 5's tables. Also please clarify whether the n=6-8 per group reported in Fig. 5's legend is per each time point (vs. pooled across them).

We have followed the suggestion of the reviewer and now have added the SEM. The number of animal used is 6-8 are per time points as indicated in the figure legend.

Reviewer #4

1. As a point of clarification regarding the statistical analyses, please denote the comparison indicated by the significance symbols in each figure. For example, in Figure 1, the meaning of # as compared to * is not defined in panel B and the reference group for these comparisons is not provided. In Figure 1e, are comparisons (*) relative to B1 or B2? In Figure 1f, does * denote a change from baseline (within a group) or a difference at a given time between 'eliminated' and 'formed' groups? Please check all figures for similar issues.

We have corrected the issue on the figure legends for each figures to make sure our statistical symbols can be understood by everyone. Changes can be found lines 1041-1043 (Figure 1b legend), lines 1095-1097 (figure 2 legend) and lines 1123-1124 (figure 3 legend). We have also now amended the figures to add the significance line to indicate between which groups the statistics were performed.

2. Please include the time point at which lesion volume was assessed.

Lesion was analyzed at 42 days post-injury. This information was added in the figure legend of supplementary figure 1 and can be found in the supplementary information file.